# Histamine synthesis and transport are coupled in axon terminals via a dual quality control system

Lei Peng[1] & Tao Wang ID [1,2,3✉]

## Abstract

**Monoamine neurotransmitters generated by de novo synthesis are rapidly transported and stored into synaptic vesicles at axon terminals. This transport is essential both for sustaining synaptic transmission and for limiting the toxic effects of monoamines. Here, synthesis of the monoamine histamine by histidine decarboxylase (HDC) and subsequent loading of histamine into synaptic vesicles are shown to be physically and functionally coupled within *Drosophila* photoreceptor terminals. This process requires HDC anchoring to synaptic vesicles via interactions with N-ethylmaleimide-sensitive fusion protein 1 (NSF1). Disassociating HDC from synaptic vesicles disrupts visual synaptic transmission and causes somatic accumulation of histamine, which leads to retinal degeneration. We further identified a proteasome degradation system mediated by the E3 ubiquitin ligase, purity of essence (POE), which clears mislocalized HDC from the soma, thus eliminating the cytotoxic effects of histamine. Taken together, our results reveal a dual mechanism for translocation and degradation of HDC that ensures restriction of histamine synthesis to axonal terminals and at the same time rapid loading into synaptic vesicles. This is crucial for sustaining neurotransmission and protecting against cytotoxic monoamines.**

**Keywords** Histamine; Synaptic Vesicle; Hdc; NSF1; POE
**Subject Categories** Membranes & Trafficking; Neuroscience

## Introduction

Rapid and efficient generation of synaptic vesicles (SVs) that contain neurotransmitters is essential for effective synaptic transmission during high-frequency stimulation (Ioannou et al, 2019; Kaeser and Regehr, 2017; Xu and Wang, 2019). Disturbing the homeostatic balance between neurotransmitter synthesis, metabolism, and transport results in diminished neuronal signaling and a wide range of neurological disorders (Behnia and Desplan, 2015). In presynaptic neurons, monoamine neurotransmitters such as dopamine and serotonin are synthesized within the cytosol. Subsequently, they are transported and stored in large dense core vesicles (LDCVs) and SVs by vesicular monoamine transporter 2 at axon terminals (Fon et al, 1997; Liu et al, 1992; Lotharius and Brundin, 2002; Nirenberg et al, 1995). Physical interactions between VMAT2 and enzymes responsible for synthesizing monoamines (e.g., tyrosine hydroxylase (TH) and aromatic amino acid decarboxylase (AADC)) are proposed to functionally couple the synthesis and vesicular loading of neurotransmitters, thus promoting the rapid and efficient replenishing of SVs to sustain synaptic transmission (Cartier et al, 2010; Parra et al, 2016). However, as a large amount of TH and AADC are not associated with synaptic vesicles, the physiological role of coupling neurotransmitter synthesis and their loading into vesicles remains unclear.

By contrast, it is clear that the dysregulation of monoamine homeostasis can lead to elevated levels of cytosolic neurotransmitters, which is pathogenic (Alter et al, 2013; Jaiswal et al, 2015). Biallelic mutations in *VMAT2* result in brain dopamine-serotonin vesicular transport disease, which is not alleviated by treatment with levodopa to increase dopamine synthesis. Mice with reduced levels of VMAT2 expression display responsive behavioral deficits and neurodegeneration of the nigrostriatal dopamine system (Caudle et al, 2007; Rilstone et al, 2013). VMAT2 overexpression protects neurons from MPTP (1-methyl-4-phenyl-1,2,3,6-tetrahydropyridine)-induced Parkinson's disease-related neurodegeneration, whereas overexpression of dopamine transporter (DAT) leads to spontaneous loss of midbrain dopamine neurons and sensitive to MPTP-induced neurotoxicity (Lohr et al, 2014; Lohr et al, 2016; Masoud et al, 2015). Moreover, dopamine is cytotoxic to neurons—dopamine accumulation in the cytosol is destructive, whereas maintaining low cytosolic concentrations of dopamine in neurons is protective (Burbulla et al, 2017; Graves et al, 2020; Mor et al, 2017; Mosharov et al, 2009). Thus, the physical and functional coupling of monoamine synthesis and their loading into synaptic vesicles may contribute to maintaining low levels of cytosolic monoamines.

The monoamine histamine is the primary mediator of anaphylaxis, but histamine also functions as a monoamine neurotransmitter, primarily in neurons of the basal ganglia. There, it modulates multiple brain functions, including sleep and wakefulness (Haas et al, 2008; Panula and Nuutinen, 2013). In both vertebrates and invertebrates, all histamine is de novo synthesized by histidine decarboxylase (Hdc), which removes a carboxyl group from histidine (Taguchi et al, 1984; Watanabe et al, 1984). Specific ablation of histamine signaling in the nervous

[1]College of Biological Sciences, China Agricultural University, Beijing, China. [2]Tsinghua Institute of Multidisciplinary Biomedical Research, Tsinghua University, Beijing 100084, China. [3]National Institute of Biological Sciences, Beijing 102206, China. ✉E-mail: wangtao1006@nibs.ac.cn

 

system leads to pathological grooming. Mutations in the *Hdc* gene result in symptoms of Tourette syndrome in both human and mice (Baldan et al, 2014; Ercan-Sencicek et al, 2010; Rapanelli et al, 2017). However, little is known about the mechanisms regulating histamine homeostasis in histaminergic neurons.

In the *Drosophila* visual system, histamine serves as the dominant neurotransmitter in conveying visual signals (Burg et al, 1993; Wang and Montell, 2007). LOVIT, a transporter responsible for loading histamine into synaptic vesicles, localizes exclusively to SVs in axonal terminals (Deshpande et al, 2020; Ioannou et al, 2019). Moreover, Both Hdc and the histidine transporter, TADR, which imports histamine precursors into the cell, are enriched in axon terminals (Han et al, 2022). Based on the co-localization of TADR, Hdc, and LOVIT, we hypothesized that histamine synthesis and the concentration of histamine in SVs function together to maintain visual transmission at high frequencies. We, therefore, uncoupled these two processes by mislocalizing Hdc and found reduced axonal histamine, disrupted visual transmission, elevated levels of cytosolic histamine, and, ultimately, neurodegeneration.

## Results

### Hdc synaptic localization determines histamine local biosynthesis

The synaptic transmission of visual information from *Drosophila* photoreceptor cells to interneurons is mediated exclusively by the neurotransmitter histamine (Wang and Montell, 2007). Hdc and LOVIT mediate histamine synthesis and the loading of histamine into SVs, respectively. They localize exclusively to the axonal region of photoreceptor neurons (lamina R1–R6 photoreceptors and medulla R7–R8 cells) (Fig. EV1A) (Han et al, 2022; Ioannou et al, 2019). Co-localization of the histamine synthesizing enzyme, Hdc, and the vesicular histamine transporter, LOVIT, suggests that the de novo synthesis of this neurotransmitter occurs locally at synaptic terminals and that the neurotransmitter is then rapidly loaded into SVs.

To test whether the synthesis of histamine in axonal terminals is required for rapid and high-frequency visual transmission, we asked whether Hdc localization to synaptic terminals is required for normal visual transmission. We first expressed mCherry-tagged Hdc in photoreceptor cells using a photoreceptor cell-specific promoter, the *trp* (*transient receptor potential*) promoter. We found that Hdc-mCherry localized to synaptic terminals, as seen with endogenous Hdc (Figs. 1A and EV1A). In contrast, GFP expressed under the control of the *trp* promoter localized to the retina, and photoreceptor axons in lamina and medulla (Fig. EV1B). Importantly, the functionality of mCherry-tagged Hdc was confirmed, as the *trp-Hdc-mCherry* transgene successfully restored normal visual transmission in *Hdc* mutant flies (Burg et al, 1993; Han et al, 2022) (Fig. 1D–F). We next fused Hdc-mCherry to the endoplasmic reticulum (ER) membrane protein EMC5 (ER Membrane Complex subunit 5), or to the SV protein synaptotagmin-1 (Syt1). These should anchor Hdc-mCherry to the soma or axon regions, respectively (Xiong et al, 2020; Zhang et al, 2002). In *Syt1-Hdc* transgene flies, all HDC localized to synapses, whereas in *emc5-Hdc* flies, Hdc-mCherry signals were dramatically reduced. This resulted

from a strict Hdc quality control pathway, as Hdc that was mislocalized to the soma was degraded (Fig. 1A). To induce Hdc activity in the soma, we expressed mCherry-tagged human Hdc (hHdc), which we hypothesized would be resistant to this endogenous quality control system, in photoreceptor cells under the *trp* promoter. By contrast to fly Hdc, hHdc levels were high and localized evenly throughout the retina, lamina, and medulla, indicating that hHdc was resistant to the Hdc quality control system and/or lacked the appropriate axonal localization signal (Fig. 1A). We then employed the anchor system for hHdc and found that Syt1-hHdc localized to synaptic terminals and that EMC5-hHdc was concentrated in the soma, as expected (Fig. 1A). Importantly, these chimeric Hdc's effectively synthesized histamine, as all of these *Hdc* or *hHdc* transgenes restored total histamine levels in *Hdc* mutants (except for the unstable EMC5-Hdc) (Fig. EV1C).

Correlated with the synaptic localization of endogenous Hdc, histamine signals in wild-type flies were detected exclusively in photoreceptor terminals of both the lamina and medulla. Similarly, when Hdc, Syt1-Hdc, or Syt1-hHdc were expressed via the *trp* promoter, they localized to photoreceptor terminals, and histamine localized there as well. By contrast, hHdc and the soma-localized EMC5-hHdc failed to concentrate histamine in synaptic terminals (Fig. 1B,C). Importantly, the localization of Hdc positively correlated with the distribution of histamine (Fig. 1C). To examine if histamine synthesis must be confined to synaptic terminals for effective neuronal transmission, we performed electroretinogram (ERG) recordings, which are extracellular recordings that measure the summed responses of all photoreceptor cells in response to light. The ON and OFF transients in ERG recordings reflect synaptic transmission from photoreceptors to laminal LMCs (large monopolar cells) neurons, whereas the sustained corneal negative response results from photoreceptor depolarization (Fig. 1D) (Wang and Montell, 2007). Hdc mutant flies lost the ERG ON and OFF transients specifically, indicating a lack of histamine. This was restored by the *trp-Hdc* transgene. Importantly, the synaptically localized Syt1-Hdc and Syt1-hHdc proteins also rescued the ERG transients, whereas the non-specifically localized hHdc and the soma-localized EMC5-hHdc failed to restore the ON and OFF transients of *Hdc* mutant flies (Fig. 1D,E). Loss of visual transmission results in blindness, which is reflected in the loss of phototactic behavior (Fig. 1F) (Behnia and Desplan, 2015). Consistent with these electrophysiology results, *Hdc* mutant flies exhibited defective phototaxis, which was restored by expressing wild-type Hdc, Syt1-Hdc, or Syt1-hHdc, but not restored by expressing hHdc or EMC5-hHdc (Fig. 1F). Taken together, these results revealed that local synthesis of histamine is required for normal synaptic transmission, which is controlled by localization of the Hdc enzyme.

### Hdc is co-transported with synaptic vesicles to axonal terminals

The localization of Hdc to synaptic terminals could be explained in two ways: (1) the local synthesis of Hdc in axonal terminals or (2) the transport of mature Hdc to synaptic terminals. We first reasoned that the enrichment of *Hdc* mRNA in photoreceptor terminals may be required for the local synthesis of Hdc. However, quantifying the ratio of *Hdc* mRNA between retina and lamina

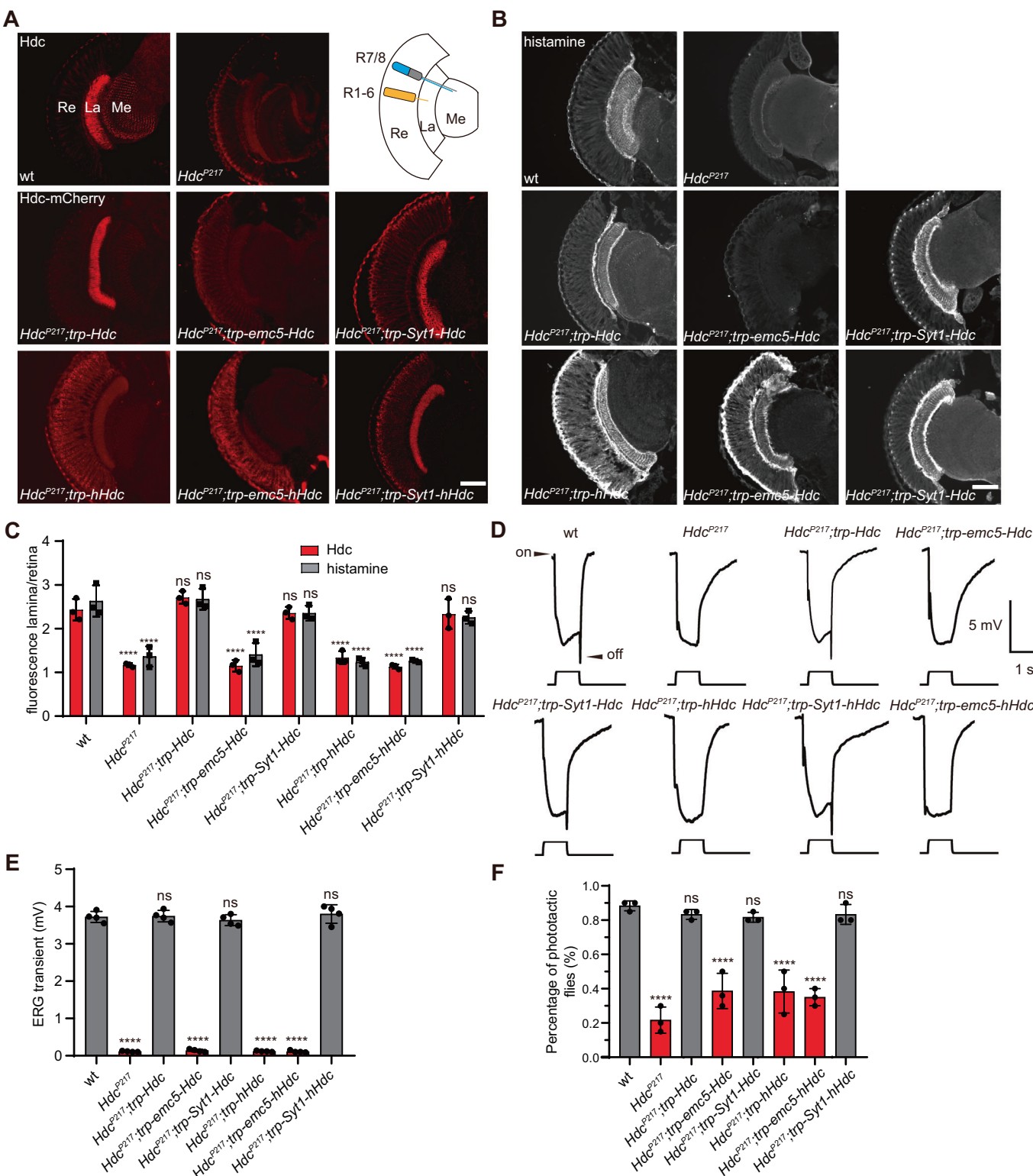

dissected from adult flies, we found that *Hdc* mRNA was enriched in retina, similar to the distribution of *hHdc* mRNA when expressed via the *trp* promoter (Fig. EV2A). Since this suggested that Hdc was not translated into axonal terminals, we reasoned that Hdc may instead be transported to synaptic terminals. Given that

endogenous Hdc colocalized with the SV marker LOVIT, we first asked if Hdc was associated with SVs. By expressing Syt1 with tandemly fused GFP and HA tags in photoreceptor cells via the combination of *UAS-Syt1-GFP-HA* and *xport-gal4* (a Gal4 line specific for photoreceptor cells), we labeled SVs with HA and GFP

◄ **Figure 1. Hdc synaptic localization is required for synaptic transmission.**

(A, B) Horizontal sections of wild-type (wt) and *Hdc* mutant flies (*Hdc^P217*) were labeled using antibodies against Hdc (A, upper) and histamine (B). *Hdc^P217* flies expressing mCherry-tagged Hdc (*Hdc^P217;trp-Hdc*), EMC5-Hdc (*Hdc^P217;trp-emc5-Hdc*), Syt1-Hdc (*Hdc^P217;trp-Syt1-Hdc*), hHdc (*Hdc^P217;trp-hHdc*), EMC5-hHdc (*Hdc^P217;trp-emc5-hHdc*), and Syt1-hHdc (*Hdc^P217;trp-Syt1-hHdc*) in photoreceptor cells were labeled using mCherry (A, middle and bottom) or histamine (B) antibodies. Scale bars, 20 μm. La lamina, Me medulla, Re retina, R1-6 R1–R6 photoreceptor cells, R7/8 R7 and R8 photoreceptor cells. (C) Quantification of fluorescence intensity ratios of Hdc-mCherry and histamine between the entire lamina and the retina based on sections shown in (A, B). Three images of each genotype were used for quantification. Hdc quantification: wt vs *Hdc^P217* ****$p < 0.0001$, wt vs *Hdc^P217;trp-Hdc* $p = 0.3770$, wt vs *Hdc^P217;trp-emc5-Hdc* ****$p < 0.0001$, wt vs *Hdc^P217;trp-Syt1-Hdc* $p = 0.9972$, wt vs *Hdc^P217;trp-hHdc* ****$p < 0.0001$, wt vs *Hdc^P217;trp-emc5-hHdc* ****$p < 0.0001$, wt vs *Hdc^P217;trp-Syt1-hHdc* $p = 0.9879$. Histamine quantification: wt vs *Hdc^P217* ****$p < 0.0001$, wt vs *Hdc^P217;trp-Hdc* $p = 0.9996$, wt vs *Hdc^P217;trp-emc5-Hdc* ****$p < 0.0001$, wt vs *Hdc^P217;trp-Syt1-Hdc* $p = 0.3889$, wt vs *Hdc^P217;trp-hHdc* ****$p < 0.0001$, wt vs *Hdc^P217;trp-emc5-hHdc* ****$p < 0.0001$, wt vs *Hdc^P217;trp-Syt1-hHdc* $p = 0.1301$. Dunnett's two-way ANOVA, mean ± sd, ns not significant. (D) Electroretinograms (ERGs) recorded from wt, *Hdc^P217* mutant flies, and *Hdc^P217* flies expressing Hdc, EMC5-Hdc, Syt1-Hdc, hHdc, Emc5-hHdc, or Syt1-hHdc. (E) Quantitative analysis of the amplitudes of ERG OFF transients shown in (D). ERG from 4 flies of each genotype were used for quantification. wt vs *Hdc^P217* ****$p < 0.0001$, wt vs *Hdc^P217;trp-Hdc* $p = 0.9998$, wt vs *Hdc^P217;trp-emc5-Hdc* ****$p < 0.0001$, wt vs *Hdc^P217;trp-Syt1-Hdc* $p = 0.9271$, wt vs *Hdc^P217;trp-hHdc* ****$p < 0.0001$, wt vs *Hdc^P217;trp-emc5-hHdc* ****$p < 0.0001$, wt vs *Hdc^P217;trp-Syt1-hHdc* $p = 0.9772$. Dunnett's one-way ANOVA, mean ± sd, ns=not significant. (F) Phototactic behaviors of the *Hdc^P217* flies expressing Hdc, EMC5-Hdc, Syt1-Hdc, hHdc, EMC5-hHdc, or Syt1-hHdc compared with wild-type flies. Three repeats were made for each group containing 20 flies. wt vs *Hdc^P217* ****$p < 0.0001$, wt vs *Hdc^P217;trp-Hdc* $p = 0.9176$, wt vs *Hdc^P217;trp-emc5-Hdc* ****$p < 0.0001$, wt vs *Hdc^P217;trp-Syt1-Hdc* $p = 0.7642$, wt vs *Hdc^P217;trp-hHdc* ****$p < 0.0001$, wt vs *Hdc^P217;trp-emc5-hHdc* ****$p < 0.0001$, wt vs *Hdc^P217;trp-Syt1-hHdc* $p = 0.9176$. Dunnett's one-way ANOVA, mean ± sd, ns not significant. Source data are available online for this figure.

tags (Fig. EV2B) (Chen et al, 2015). Syt1-labeled SVs were purified via immunoprecipitation (IP) with anti-HA magnetic beads. We confirmed the efficiency of purification by enrichment of an SV marker (CSP) and the absence of an ER protein (CNX99), a mitochondrial protein (TOM20), and a cytosolic protein (Tan). Importantly, Hdc was co-purified with SVs (Fig. 2A).

Since Hdc was associated with SVs, we hypothesized that Hdc is co-transported to axonal terminals with synaptic vesicles. The kinesin motor is responsible for the anterograde trafficking of vesicles along the microtubule filament to neuronal terminals. Knocking down *unc-104* or *Khc*, which encodes kinesin motor proteins required for SV transport, reduced levels of the SV transporter LOVIT in the lamina (Figs. EV2C and 2C). As seen with LOVIT, Hdc levels were also reduced in synaptic terminals by knocking down UNC-104 or Khc. By contrast, levels of Tan, a cytosolic N-β-alanyl-dopamine hydrolase that generates histamine from carcinine, were unaffected (Fig. 2B,C). These data suggest that Hdc is transported with SVs to axonal terminals.

## Hdc must bind NSF1 to localize to axons

Hdc is a cytosolic protein that lacks transmembrane hydrophobic regions, and thus alone could not associate with SV membranes. We reasoned that Hdc may bind membrane proteins on the surface of SVs. To explore this possibility, we performed an immunoprecipitation assay with the RFP-Trap using flies expressing mCherry-tagged Hdc, followed by mass spectra identification. This approach identified four proteins that bound Hdc-mCherry but not mCherry alone (Table EV1). Among these four candidates, NSF1 (*N*-ethylmaleimide-Sensitive Factor 1) is an SV membrane protein required for the maintenance of neurotransmitter release (Babcock et al, 2004; Tolar and Pallanck, 1998). Consistent with its function in SVs, we confirmed that GFP-tagged NSF1 localized to axonal terminals with Hdc (Fig. EV3A). Further genetic analysis showed that among the four candidates, knocking down NSF1 using two distinct RNAi lines in photoreceptor cells reduced Hdc levels in synaptic terminals. By contrast, knocking down the other 3 genes did not affect Hdc levels in axonal terminals (Figs. 3A and EV3B,C). In line with the mislocalization of Hdc, histamine was not concentrated in the synaptic terminals of flies expressing *NSF1^RNAi* (Fig. EV3D).

We next expressed mCherry-tagged Hdc and MYC-tagged NSF1 in S2 cells, and found that NSF1 efficiently co-immunoprecipitated with Hdc (Fig. 3B). As a member of the AAA family of ATPases, NSF1 ATPase function plays a crucial role in membrane fusion processes (Littleton et al, 1998; Müller et al, 1999; Schweizer et al, 1998; Söllner et al, 1993). To determine whether the ATPase activity of NSF1 is essential for its interaction with Hdc, we expressed mCherry-tagged Hdc along with MYC-tagged NSF1^G274E, a mutation that disrupts ATPase activity of NSF1, in S2 cells. Importantly, we found that the NSF1^G274E mutation has no effects on the binding ability of NSF1 and Hdc, suggesting that NSF1's role in mediating axonal trafficking of Hdc is independent of its ATPase activity (Müller et al, 1999) (Fig. EV3E). Furthermore, in vivo, co-immunoprecipitation experiments using transgenic flies co-expressing NSF1-GFP and Hdc-mCherry in photoreceptor cells confirmed efficient interaction between NSF1 and Hdc (Fig. EV3F). Of importance, knocking down NSF1 did not affect LOVIT localization, suggesting that NSF1 is not involved in trafficking SVs within axons (Fig. 3A).

To further characterize the interaction between Hdc and NSF1, we split Hdc into its N-terminal region (Hdc-N-mCherry), its enzyme domain (Hdc-M-mCherry), and its C-terminal region (Hdc-C-mCherry), tagging each with mCherry. By co-expressing these truncated Hdc's with MYC-tagged NSF1 in S2 cells, we found that NSF1 efficiently co-immunoprecipitated only with Hdc-N-mCherry. Further, a version of Hdc lacking its N-terminal 34 amino acids (Hdc-dN-mCherry) did not interact with NSF1 (Fig. 3C). We next expressed Hdc-N-mCherry, Hdc-dN-mCherry, Hdc-M-mCherry, or Hdc-C-mCherry in photoreceptor cells using the *trp* promoter and examined the localization of these truncated Hdc's. Consistent with their ability to interact with NSF1, the 34 N-terminal amino acids of Hdc were sufficient to anchor mCherry to synaptic terminals. By contrast, versions of Hdc without the N-terminal did not enrich mCherry in neuronal terminals (Figs. 3D and EV3G). Notably, Hdc-dN-mCherry was associated with lower overall Hdc signals, supporting the notion that Hdc that was mislocalized to the soma was degraded. Consistent with this, Hdc-dN-mCherry did not restore ERG transients in *Hdc* mutant flies. However, the Syt1-Hdc-dN-mCherry protein, which resulted in the localization of the N-terminal-deleted Hdc to SVs, rescued visual transmission defects (Figs. 3D and EV3H). These results revealed that Hdc is transported with SVs by binding NSF1, and that this is critical for synaptic transmission.

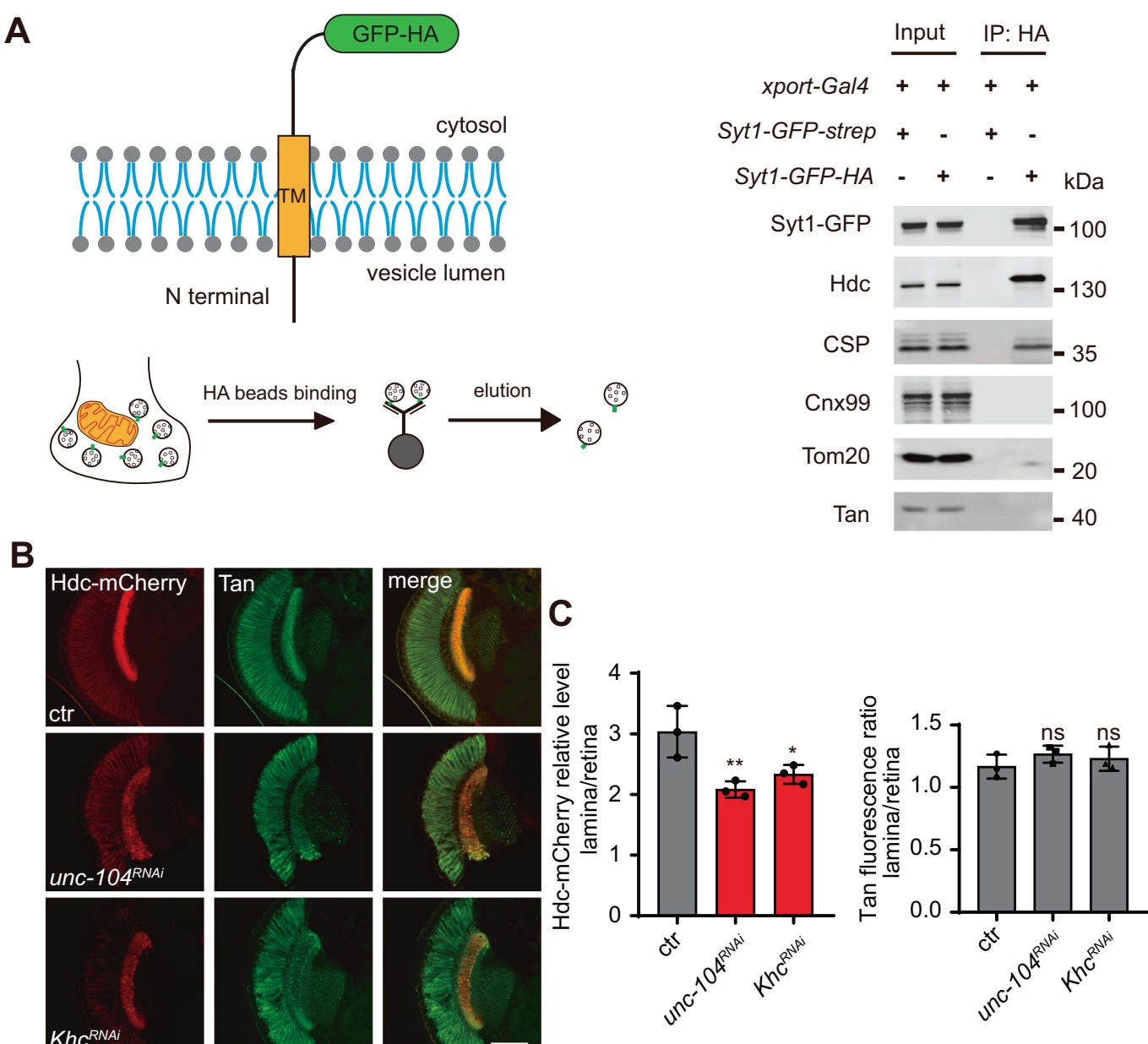

**Figure 2. Hdc is co-transported with synaptic vesicles to axonal terminals.**

(A) Schematic diagram illustrating the isolation of synaptic vesicles (SVs) from flies expressing HA and GFP-tagged Syt1 in photoreceptor cells (left). Immunoblot analysis for Syt1- GFP, Hdc, CSP (SV protein), Cnx99 (ER protein), Tom20 (mitochondrial protein), and Tan (cytosolic protein) from total lysates (left) and isolated SVs (right). Lysates were prepared from flies expressing Syt1-GFP-HA, and control flies expressing Syt1-GFP-strep in photoreceptor cells driven by *xport-Gal4*. (B) Sections of *trp-Hdc-mCherry* flies expressing *GFP^RNAi* (control), *unc-104^RNAi* (*GMR>unc-104^RNAi*, *GMR-Gal4/UAS-unc-104^RNAi*), or *Khc^RNAi* (*GMR > Khc^RNAi*, *GMR-Gal4/UAS-Khc^RNAi*) in the eye were labeled with antibodies against mCherry (red) and Tan (green). Scale bars, 20 μm. (C) Quantitative ratios of Hdc-mCherry and Tan levels between lamina and retina of *trp-Hdc-mCherry* flies expressing *GFP^RNAi* (control), *unc-104^RNAi*, or *Khc^RNAi*. Sections of 3 *Drosophila* heads were used for quantification. Hdc-mCherry level quantification: *GFP^RNAi* vs *unc-104^RNAi* **$p = 0.0094$, *GFP^RNAi* vs *Khc^RNAi* *$p = 0.0379$. Tan level quantification: *GFP^RNAi* vs *unc-104^RNAi* $p = 0.3407$, *GFP^RNAi* vs *Khc^RNAi* $p = 0.6079$. Dunnett's one-way ANOVA, mean ± sd, ns not significant. Source data are available online for this figure.

## Hdc is degraded by the ubiquitin-proteasome system in the soma

When Hdc was not transported to axons, as seen in flies expressing EMC5-Hdc or Hdc-dN, protein levels were greatly reduced (Fig. 4A). By contrast, mRNA levels for these Hdc variants remained unchanged (Fig. EV4A). We therefore hypothesized that

there is a quality control system in photoreceptors to degrade Hdc that is retained in the cell body. In eukaryotic cells, two major pathways mediate protein degradation, the ubiquitin-proteasome system (UPS) and lysosomal proteolysis. Interruption of UPS by knocking down the proteasome subunits α5 (*Pros-α5*) and β1 (*Pros-β1*) increased levels of Hdc-mCherry in *trp-Hdc-mCherry* flies. Hdc levels were not affected when autophagy was disrupted by knocking

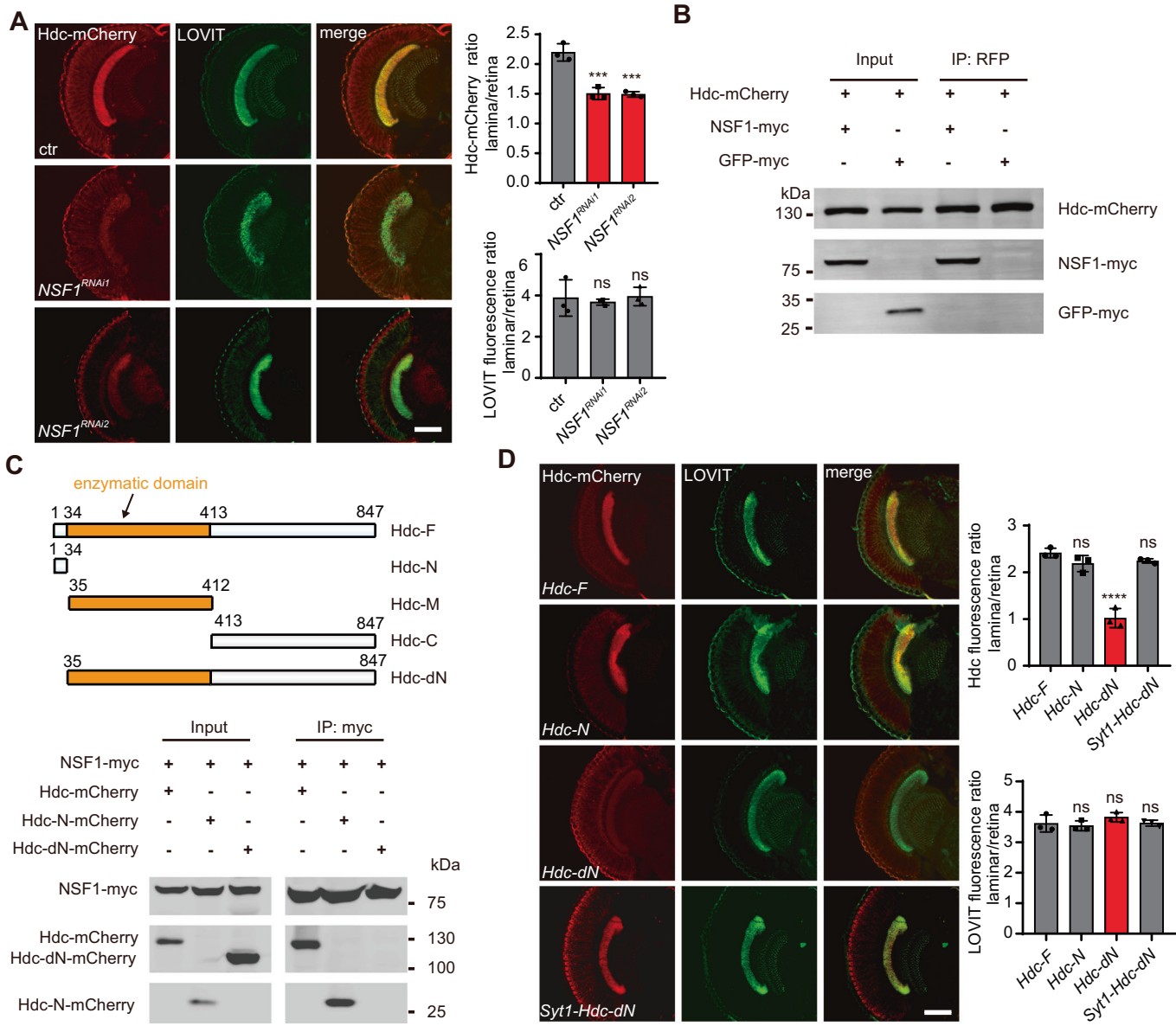

**Figure 3. Hdc binds NSF1, maintaining its axonal localization.**

(A) Horizontal head sections of *trp-Hdc-mCherry* flies with eye-specific expression of *GFP^RNAi* (control), *NSF1^RNAi1* (*GMR>NSF1^RNAi1*, *GMR-Gal4/UAS-NSF1^RNAi1*) and *NSF1^RNA2* (*GMR>NSF1^RNA2*, *GMR-Gal4/UAS-NSF1^RNAi2*) were immunostained with antibodies against mCherry (red) and LOVIT (green). Scale bars, 20 μm. Ratios of Hdc-mCherry signals between the lamina and retina were quantified. ctr vs *NSF1^RNAi1* ***p = 0.0004, ctr vs *NSF1^RNAi2* ***p = 0.0003. Ratios of LOVIT fluorescence between the lamina and retina were quantified. ctr vs *NSF1^RNAi1* p = 0.8769, ctr vs *NSF1^RNAi2* p = 0.9840. Dunnett's one-way ANOVA, n = 3, mean ± sd, ns=not significant. (B) Hdc physically interacts with NSF1. Hdc-mCherry was co-expressed with NSF1-Myc or GFP-Myc in S2 cells. Cell lysates were immunoprecipitated with anti-mCherry beads, and blotted against mCherry and Myc. (C) Full-length Hdc (Hdc_{1-847}) was truncated into Hdc-N (Hdc_{1-34}), Hdc-M (enzymatic domain, Hdc_{35-412}), Hdc-C (Hdc_{413-847}), and Hdc-dN (Hdc_{35-847}). NSF1-Myc was co-expressed with Hdc-N-mCherry and Hdc-dN-mCherry in S2 cells, immunoprecipitated with Myc beads, and blotted against Myc and mCherry antibodies. (D) Sections of fly heads expressing Hdc-F (*trp-Hdc-F-mCherry*), Hdc-N (*trp-Hdc-N-mCherry*), Hdc-dN (*trp-Hdc-dN-mCherry*), or Syt1-Hdc-dN (*trp-Syt1-Hdc-dN-mCherry*) in photoreceptor cells were stained with mCherry and LOVIT antibodies. Scale bars, 20 μm. Ratios of Hdc-mCherry signals between the lamina and retina were quantified. Hdc-F vs Hdc-N p = 0.2055, Hdc-F vs Hdc-dN ****p < 0.0001, Hdc-F vs Syt1-Hdc-dN p = 0.3924. Ratios of LOVIT signals between the lamina and retina were quantified. Hdc-F vs Hdc-N p = 0.9201, Hdc-F vs Hdc-dN p = 0.4407, Hdc-F vs Syt1-Hdc-dN p = 0.9999. n = 3. Dunnett's one-way ANOVA, mean ± sd, ns not significant. Source data are available online for this figure.

down the core autophagy genes *atg7* and *atg8a* (Fig. 4B). We further labeled retinas for Hdc-mCherry, and found that mCherry signals increased upon induction of *Pros-α5^RNAi* and *Pros-β1^RNAi* (Fig. EV4B,C).

E3 ubiquitin ligases are key factors in determining the specificity of protein substrates in UPS, and may be responsible for Hdc quality control. In the *Drosophila* genome, there are ~200 genes that encode E3 ubiquitin ligases, and ~88 of these genes are

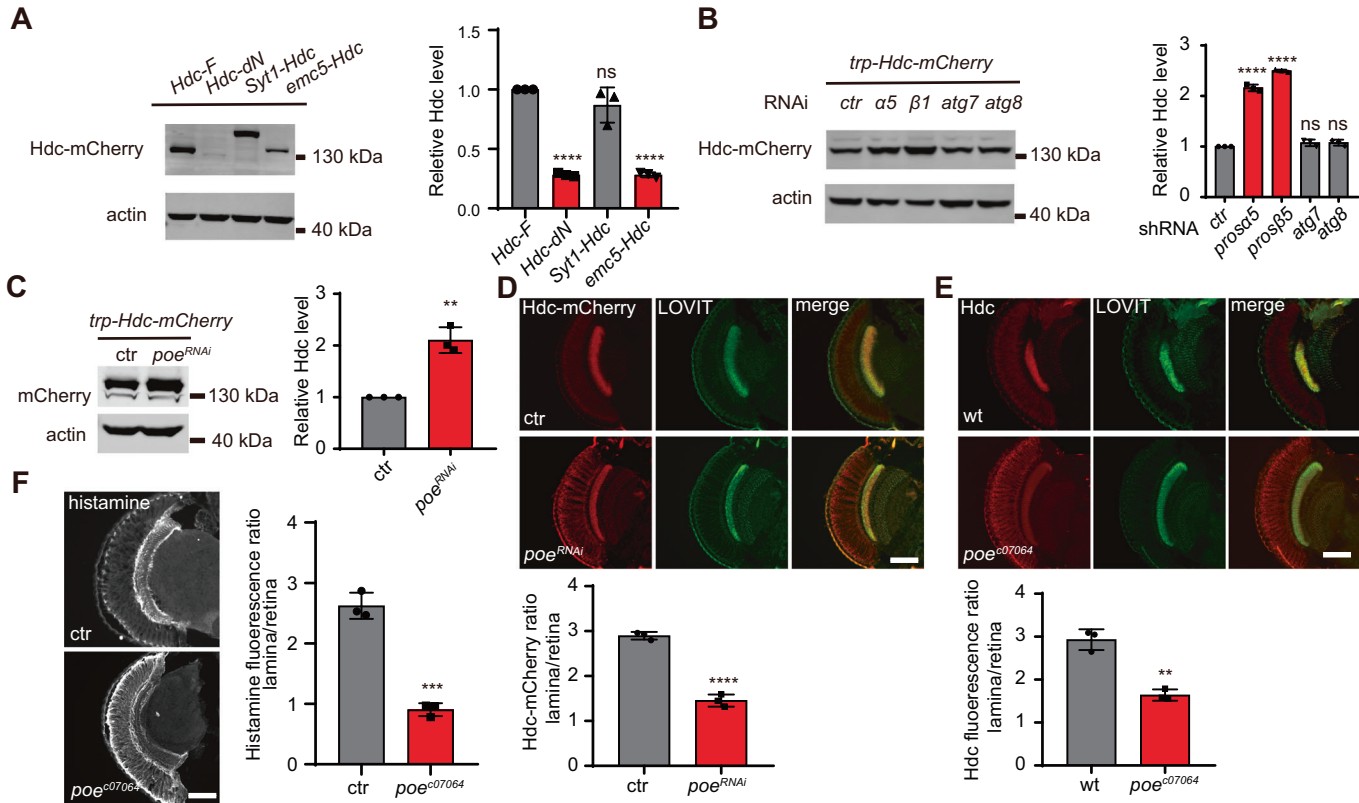

**Figure 4. Somatic Hdc is degraded by a POE-dependent ubiquitin-proteasome system.**

(A) Western blot analysis of Hdc-mCherry protein levels from flies expressing Hdc-F (*trp-Hdc-mCherry*), Hdc-dN (*trp-Hdc-dN-mCherry*), Syt1-Hdc (*trp-Syt1-Hdc-mCherry*), or Emc5-Hdc (*trp-emc5-Hdc-mCherry*) using antibodies against mCherry. Actin was used as an internal loading control. Hdc-F vs Hdc-dN ****$p < 0.0001$, Hdc-F vs Syt1-Hdc $p = 0.1538$, Hdc-F vs emc5-Hdc ****$p < 0.0001$. Dunnett's one-way ANOVA, $n = 3$, mean ± sd, ns not significant. (B) Lysates from *trp-Hdc-mCherry* flies expressing *RNAis* against proteasome subunits (*prosα5^RNAi^*, *prosβ1^RNAi^*) and autophagy-related genes (*atg7^RNAi^*, *atg8^RNAi^*) driven by the *GMR-Gal4* driver were blotted with mCherry and actin antibodies. Relative Hdc protein levels normalized to actin levels were quantified. Three blots were used for quantification. ctr vs *prosα5^RNAi^* ****$p < 0.0001$, ctr vs *prosβ1^RNAi^* ****$p < 0.0001$, ctr vs *atg7^RNAi^* $p = 0.2771$, ctr vs *atg8^RNAi^* $p = 0.2464$. Dunnett's one-way ANOVA, $n = 3$, mean ± sd, ns not significant. (C) Western blot comparing Hdc protein levels in *trp-Hdc-mCherry* flies knocking down *poe* (*GMR>poe^RNAi^*) with control (*GMR>GFP^RNAi^*). Hdc levels were normalized to actin. ctr vs *poe^RNAi^* **$p = 0.0016$. Unpaired $t$-test, $n = 3$, mean ± sd. (D) Immunostaining comparing Hdc-mCherry levels in *trp-Hdc-mCherry* flies knocking down *poe* (*GMR>poe^RNAi^*) with control (*GMR>GFP^RNAi^*). ctr vs *poe^RNAi^* ****$p < 0.0001$. Unpaired $t$-test, $n = 3$, mean ± sd. (E) Immunostaining of flies with *poe^c07064^* homozygous mutant eye with anti-Hdc (red) and anti-LOVIT (green) antibodies. Relative laminal Hdc levels against retina levels were quantified. ctr vs *poe^c07064^* **$p = 0.0013$. Unpaired $t$-test, $n = 3$, mean ± sd. (F) Head horizontal sections from wild-type and *poe^c07064^* mutant flies were immunolabeled with histamine, and the ratios of histamine signal intensity between lamina and retina were quantified. ctr vs *poe^c07064^* ***$p = 0.0002$. Unpaired $t$-test, $n = 3$, mean ± sd. Scale bars, 20 μm. Source data are available online for this figure.

expressed in the fly eye (Du et al, 2011; Xu and Wang, 2016). We knocked down each candidate E3 gene via the eye-specific expression of *RNAi* and directly examined Hdc-mCherry levels using a fluorescence stereo microscope (Table EV2). We found that knockdown of *purity of essence* (*poe*), which encodes an unclassified E3 ubiquitin ligase, increased total Hdc-mCherry levels (Fig. 4C). Further, Hdc-mCherry levels were elevated in the soma upon knocking down *poe* in *trp-Hdc-mCherry* flies (Fig. 4D). We then obtained a PiggyBac insertion line, *poe^c07064^*, which contains an insertion within the coding region of the first exon of the *poe* gene and is presumably a null allele. Since *poe^c07064^* is a lethal allele, we used mitotic recombination (*ey-flp/hid* system) to generate homozygous *poe^c07064^* mutant eyes in an otherwise heterozygous body (Zhao et al, 2015b). These flies exhibited an accumulation of endogenous Hdc in the retina, confirming that POE is required for degradation of mislocalized Hdc (Fig. 4E). Further, consistent with the distribution of Hdc proteins, histamine accumulated in the cell

bodies of photoreceptor cells in *poe^c07064^* mutant flies (Fig. 4E). These results revealed that soma-retained Hdc is specifically degraded by a POE-mediated UPS pathway.

## Failure of Hdc quality control causes retinal degeneration

Given that the distribution of neurotransmitters is tightly regulated, and that histamine is found exclusively in photoreceptor axons, we next asked what effect the mis-accumulation of histamine in the soma had on neuronal integrity. As exogenous hHdc is resistant to the endogenous degradation machinery, we first examined *trp-hHdc* flies, in which histamine is detected in retina. In wild-type Hdc-expressing flies (*trp-Hdc*), seven rhabdomeres were detected regardless of age. However, aged flies (20 days) expressing hHdc (*trp-hHdc*) exhibited severe loss of rhabdomeres and photoreceptor cells (Fig. 5A,B). By contrast, expressing an enzyme-dead hHdc variant, hHdc^K305G^, which does not generate histamine, did not

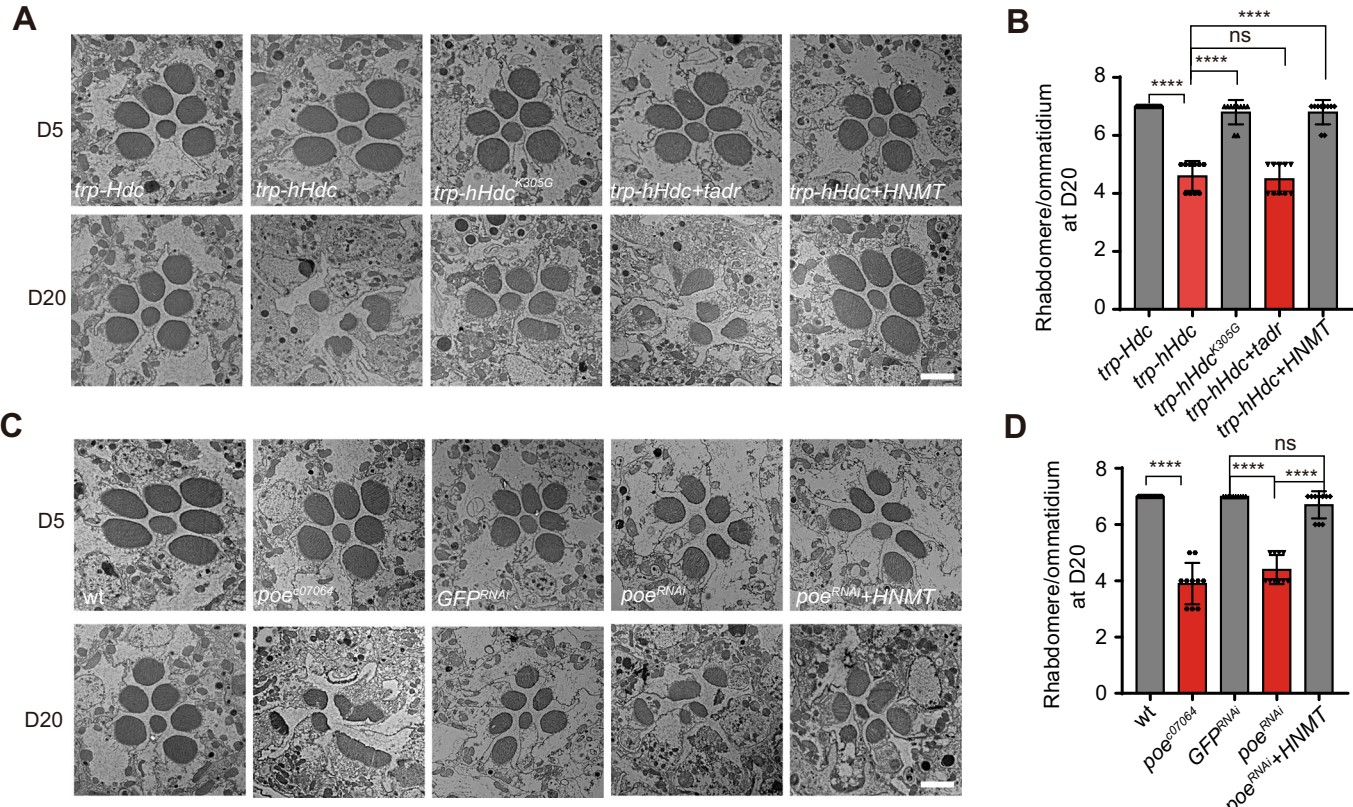

**Figure 5. Disrupting Hdc quality control machinery leads to retinal degeneration.**

(A) TEM images of eye sections from 1-day-old (upper) or 20-day-old (lower) flies expressing Hdc (*trp-Hdc*), hHdc (*trp-hHdc*), or hHdc$^{K305G}$ (*trp-hHdc$^{K305G}$*), and flies co-expressing hHdc with TADR (*trp-hHdc+tadr, trp-hHdc GMR-Gal4/UAS-tadr*) or HNMT (*trp-hHdc + HNMT, trp-hHdc GMR-Gal4/UAS-HNMT*). Scale bar, 2 μm; R rhabdomere. (B) Quantification of the mean number of rhabdomeres per ommatidium in (A). *trp-Hdc* vs *trp-hHdc* ****$p < 0.0001$, *trp-hHdc* vs *trp-hHdc$^{K305G}$* ****$p < 0.0001$, *trp-hHdc* vs *trp-hHdc+tadr* $p = 0.9841$, *trp-hHdc* vs *trp-hHdc + HNMT* ****$p < 0.0001$. Tukey's one-way ANOVA, $n = 10$, mean ± sd, ns not significant. (C) TEM images of eye sections from wild-type, *poe$^{c07064}$*, *GFP$^{RNAi}$* (*GMR-gal4/UAS-GFP*) and *poe*-knocking down flies expressing GFP (*poe$^{RNAi}$, GMR-gal4 UAS-poe$^{RNAi}$/UAS-GFP*) or HNMT (*poe$^{RNAi}$ + HNMT, GMR-gal4 UAS-poe$^{RNAi}$/UAS-HNMT*) for 1- and 20-day old flies. Scale bar, 2 μm. (D) Quantification of the number of rhabdomeres per ommatidium in (C). Quantifications were based on 10 ommatidia from three TEM sections for each genotype. All flies were raised under a 12-h light/12-h dark cycles. wt vs *poe$^{c07064}$* ****$p < 0.0001$, *GFP$^{RNAi}$* vs *poe$^{RNAi}$* ****$p < 0.0001$, *poe$^{RNAi}$* vs *poe$^{RNAi}$ + HNMT* ****$p < 0.0001$, *GFP$^{RNAi}$* vs *poe$^{RNAi}$ + HNMT*, $p = 0.5882$. Tukey's one-way ANOVA, mean ± sd, ns not significant. Source data are available online for this figure.

cause photoreceptor degeneration (Figs. 5A,B and EV5A,B) (Komori et al, 2012). Since this retinal degeneration depended on Hdc enzymatic activity in the cell body, we reasoned that depletion of histidine (an essential amino acid) or accumulation of histamine in the cytosol might be causes. To distinguish between these possibilities, we overexpressed the histidine transporter, TADR, to supply more histidine, or histamine N-methyltransferase (HNMT), which is an enzyme that degrades histamine. HNMT expression suppressed retinal degeneration associated with hHdc, whereas TADR overexpression had no effect (Fig. 5A,B). Similarly, the retinal accumulation of histamine caused photoreceptor cell degeneration in *lovit¹* mutant flies (Xu and Wang, 2019) (Fig. EV5C).

We then disrupted the Hdc quality control system by knocking down *poe*. *poe$^{RNAi}$* flies exhibited retinal degeneration and the accumulation of Hdc and histamine in photoreceptor cell bodies (Fig. 5C,D). Reducing histamine levels by expressing HNMT suppressed retinal degeneration in *poe$^{RNAi}$* flies (Fig. 5C,D). Finally, we confirmed this retinal degeneration by examining homozygous

*poe$^{c07064}$* mutant eyes, illustrating the physiological significance of restricting Hdc activity to axonal terminals (Fig. 5C,D).

## Discussion

This study answers an important question in neuroscience: what is the physiological role of functionally coupling neurotransmitter synthesis and vesicular loading? The data presented here will formulate better models regarding the coupling of histamine synthesis and transport into SVs. After Hdc is translated in the soma, the enzyme binds NSF1 on the surface of SVs and is transported to axon terminals through anterograde axonal trafficking. At the terminal, Hdc generates histamine, which is subjected to LOVIT-mediated loading into SVs. This is all critical for sustaining photoreceptor synaptic transmission (Fig. 6). Hdc that remains in the soma is degraded by a POE-dependent UPS pathway, thereby limiting the cytotoxic effects of histamine. Uncoupling these two processes (histamine generation and loading

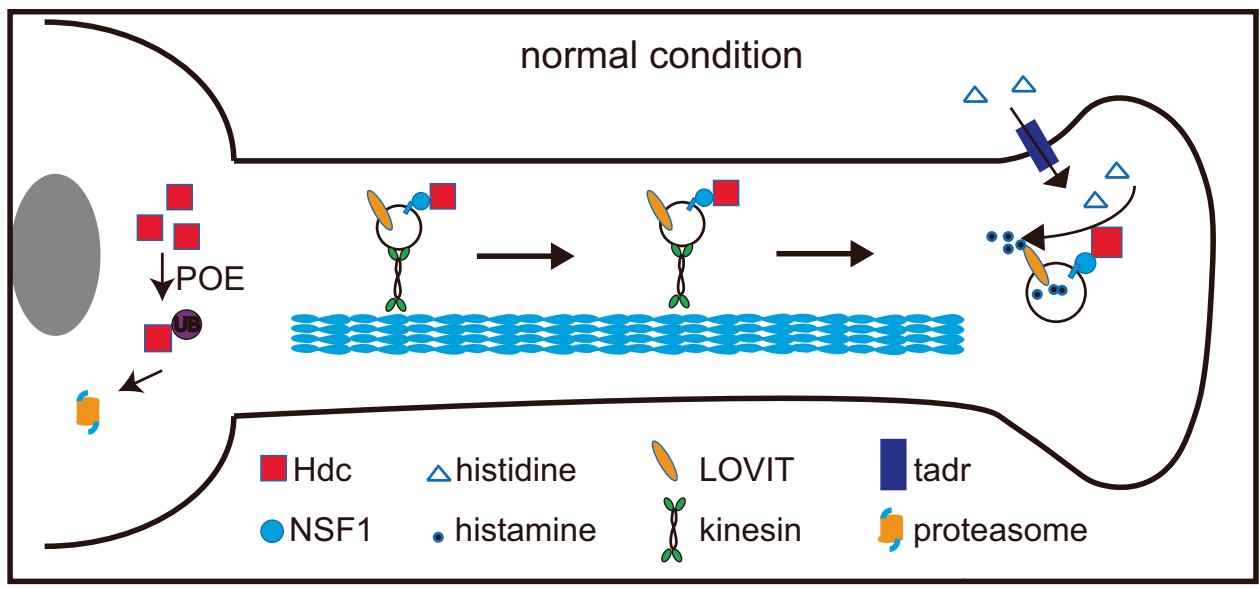

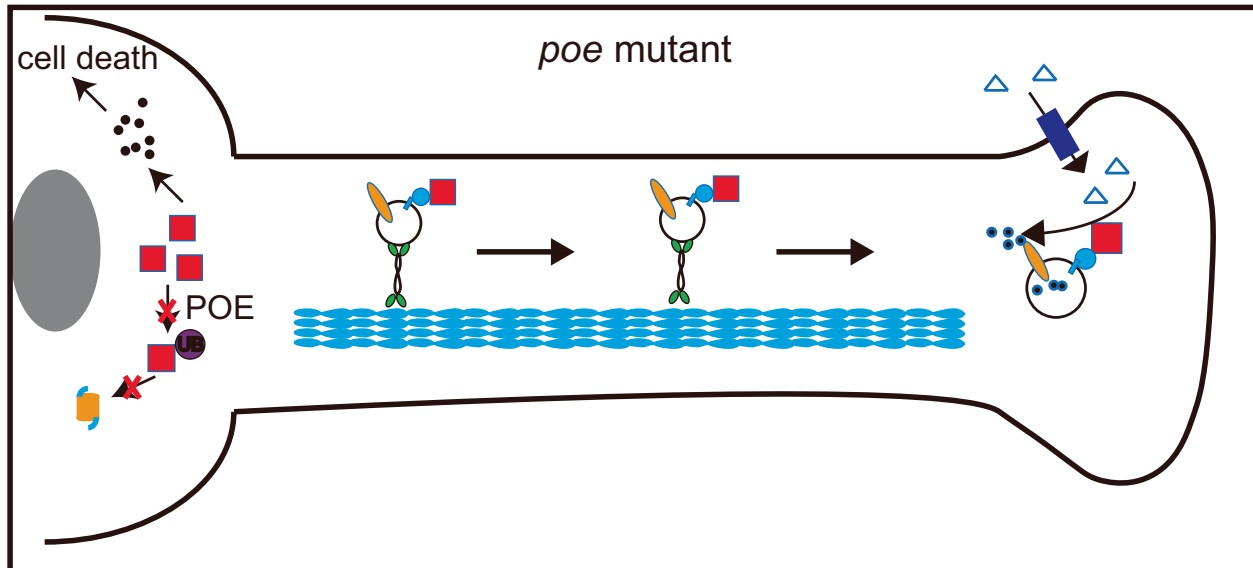

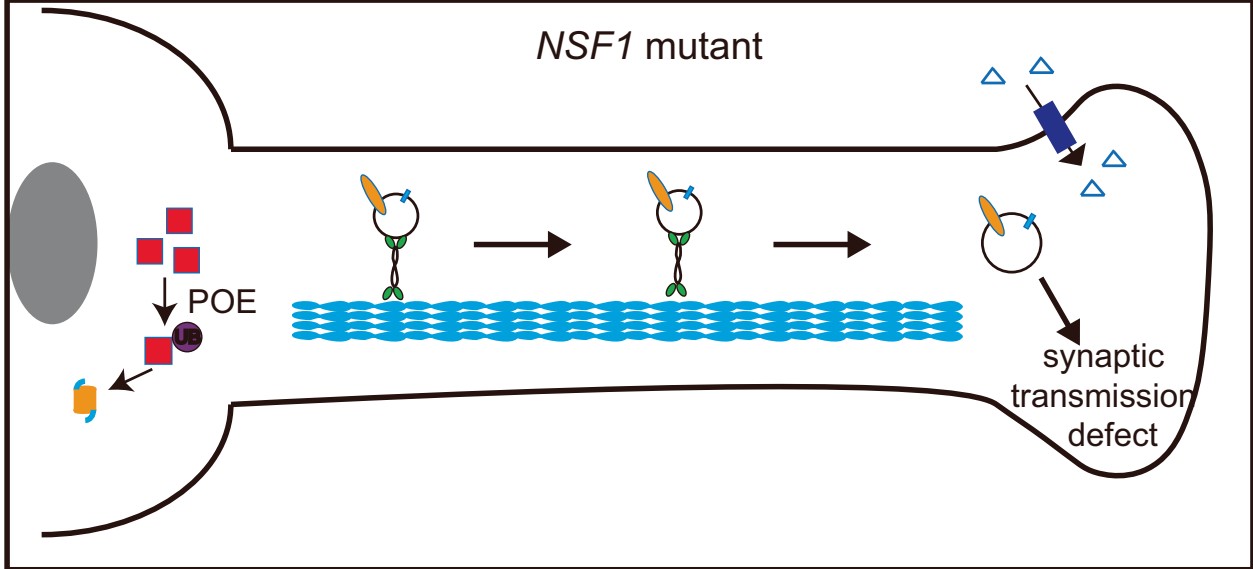

**Figure 6.  Working models for the coupling of histamine synthesis and transport.**

Newly translated Hdc binds to NSF1 and is co-transported with synaptic vesicles to synaptic terminals. Histidine is imported into presynaptic terminals by TADR, where it is converted to histamine by Hdc localized to synaptic vesicles. Adjacent LOVIT transporters pack the newly generated histamine into synaptic vesicles for visual transmission. Hdc that fails to translocate to synaptic terminals is subjected to proteasomal degradation mediated by the E3 ubiquitin ligase, POE. Disruption of Hdc trafficking to axonal terminals uncouples histamine synthesis and LOVIT-mediated transport, leading to the lack of histamine in synaptic vesicles and the loss of visual transmission. Blocking POE-mediated Hdc degradation leads to Hdc and histamine accumulation in the soma and cell death.

into SVs) through loss of POE, mislocalization of Hdc, or loss of LOVIT, disrupted visual transmission and resulted in photoreceptor cell neurodegeneration.

Under normal conditions, monoamines such as dopamine and serotine are primarily synthesized and transported to LDCVs in the soma for storage. Subsequently, they are delivered to SVs at synaptic terminals (Cartier et al, 2010; Walther et al, 2003). The finding that VMAT2 forms a complex with TH and AADC, suggests that this complex may accelerate dopamine transport into vesicles (Cartier et al, 2010; Parra et al, 2016). However, since we lack an in vivo system to specifically disrupt this physical interaction, the physiological function of this interaction remains unknown. Moreover, despite the fact that TH and AADC physically interact with VAMT2, most enzymes required for dopamine synthesis are free in the cytosol, bringing into question the physiological role of coupling neurotransmitter synthesis and vesicular loading. In photoreceptor neurons, Hdc does not directly interact with LOVIT, but it does associate with SVs. The small size and high density of SVs within synaptic terminals bring Hdc and LOVIT close together, coupling these two activities (Kaeser and Regehr, 2017). Similarly, genetically anchoring Hdc to SVs restores axonal histamine and thus visual responses in *Hdc* mutants. By contrast, the separation of Hdc and SVs by anchoring Hdc to the ER membrane failed to concentrate histamine in synaptic terminals, and thus prevented Hdc from playing a role in vision. We further demonstrated that Hdc is transported and associated with SVs by interacting with NSF1, which is targeted to axonal terminals by attaching to SVs. First, Hdc co-immunoprecipitated with NSF1, which binds and forms a complex with SNAREs in presynaptic terminals. The ATPase activity of NSF1 is essential for the fusion of SVs with the plasma membrane, but it is not necessary for its binding with Hdc (Block et al, 1988; Müller et al, 1999; Schweizer et al, 1998; Söllner et al, 1993; Tolar and Pallanck, 1998; Zhao et al, 2015a). Second, Hdc interacts with NSF1 via its N-terminal domain, which is both necessary and sufficient for the axonal localization of Hdc. Third, the knockdown of NSF1 results in reduced levels of axonal Hdc and, consequently, lower histamine levels within axonal terminals. This contrasts with the situation where disrupting SV fusion with the plasma membrane in *NSF1* mutant neurons does not change the number of SVs, and would not affect neurotransmitter concentration (Kawasaki et al, 1998). Importantly, LOVIT signals remain unaffected by NSF1 knockdown, providing strong evidence for NSF1's specific role in the proper localization of Hdc.

Rapid delivery of monoamines to SVs right after synthesis is required both to maintain neuronal activity and to prevent the cytosolic accumulation of neurotransmitters, which can interfere with neuronal functions and even cause cell death. For example, dopamine is cytotoxic to cultured cells (neurons in particular) through the induction of oxidative stress (Linert et al, 1996; Meiser et al, 2013; Segura-Aguilar et al, 2014). In Parkinson's disease, dysregulation of dopamine homeostasis results in the accumulation of dopamine, which in turn leads to the buildup of autoxidation

products, damage to mitochondria and lysosomes, and eventually, cell death of nigrostriatal neurons (Burbulla et al, 2017; Mor et al, 2017). L-DOPA is the established therapy for Parkinson's disease. However, long-term administration of the dopamine precursor often leads to dyskinesia, highlighting that both dopamine deficiency and excessive cytosolic dopamine can contribute to the pathogenesis. This dual effect can lead to the degeneration of nigral neurons in Parkinson's disease (Alter et al, 2013; Turcano et al, 2018). Unlike nigrostriatal degeneration, no cell death was detected in mutants with reduced levels of histamine, such as *Hdc* and *tadr* mutant*s* (Han et al, 2022). However, cytosolic histamine was neurotoxic to photoreceptor cells, as elevation of somatic histamine (by expressing ectopic human Hdc or by blocking the histamine vesicular transporter LOVIT) triggered photoreceptor degeneration. Consistent with the *lovit* mutation, mislocalization of Hdc to the soma resulted in the local accumulation of histamine and neuronal degeneration. Further, expressing HNMT, a mammalian enzyme that degrades histamine, suppressed the retinal degeneration associated with mislocalized Hdc. Therefore, cytosolic histamine is toxic to photoreceptor neurons, and the accumulation of free histamine triggers neuronal degeneration.

Loss-of-function mutations in the *Hdc* locus is a genetic cause of Tourette syndrome, a neurological disorder characterized by repetitive and unwanted movements, and functional impairment of histaminergic neurons is associated with multiple neurodegenerative diseases (Baldan et al, 2014; Ercan-Sencicek et al, 2010; Panula and Nuutinen, 2013). Histamine levels have also been correlated with neuroinflammation, triggering microglial phagocytosis and the production of reactive oxygen species (ROS) both in the intro and in vivo (Abdulrazzaq et al, 2022; Rocha et al, 2016). Because mammalian histaminergic neurons have not been well characterized, the *Drosophila* visual system provides an excellent in vivo model for studying how the loss of histamine homeostasis results in neurological diseases (Han et al, 2022; Wang and Montell, 2007). This is illustrated by the fact that mutations in the *Hdc*, *tadr*, and *lovit* genes reduced histamine levels and impaired synaptic transmission but did not damage the neurons. By contrast, free cytosolic histamine resulted in cell death. Since localization of histamine to the soma is harmful, we show here that cells use a protein degradation system involving the E3 ligase, POE, to degrade mislocalized Hdc and reduce cytosolic histamine. Loss of POE or disruption of the proteasome increased total Hdc levels, resulting in Hdc mislocalization to the soma. Further, disruption of this Hdc clearance pathway through mutations in *poe* led to photoreceptor degeneration, and promoting degradation of histamine by introducing HNMT prevented retinal degeneration in *poe* mutants. This finding that cytosolic histamine is cytotoxicity could help explain histamine-triggered neuroinflammation.

In addition to the de novo synthesis pathway, histamine is also recycled to help maintain histamine levels in photoreceptors

(Chaturvedi et al, 2014; Han et al, 2017). Epithelial glia takes up histamine and inactivates it to carcinine. This carcinine is then transported back into photoreceptors by CarT (carcinine-specific transporter), where *N*-β-alanyl-dopamine hydrolyzes (Tan) hydrolyzes carcinine back into histamine to restore the neurotransmitter pool (Deshpande et al, 2020). Although Tan is cytosolic and not restricted to synaptic terminals like Hdc (Borycz et al, 2002; Cartier et al, 2010; Wagner et al, 2007), CarT is found exclusively in the plasma membrane of synaptic terminals in photoreceptor neurons. Thus, carcinine is rapidly imported into axonal terminals from the synaptic cleft (Chaturvedi et al, 2016; Desplan et al, 2015; Stenesen et al, 2015). The compact space of the axonal terminal and the high density of SVs prevents diffusion of carcinine away from the synapse, functionally coupling Tan activity and the LOVIT transporter. In this way, Tan activity is confined to the synapse (and absent in the soma), protecting neurons from histamine cytotoxicity.

In this study, we discovered a double-regulation system that couples histamine synthesis and its transport to SVs to maintain neurotransmitter homeostasis. The precursor of the neurotransmitter histidine is taken up into presynaptic terminals by the TADR transporter. Histamine is then locally synthesized in the presynaptic terminal by SV-associated Hdc and rapidly transported to SVs by nearby LOVIT transporters. Thus, histamine synthesis, loading, and release occur synergistically within axonal terminals to meet the demands of phototransduction, which is the fastest known G-protein-coupled signaling cascade (Montell, 2012; Wang and Montell, 2007). Hdc that is mislocalized to the soma is degraded by a POE-mediated proteasome ubiquitin system to prevent histamine toxicity. As monoamine toxicity is also observed in the mammalian CNS, similar mechanisms for limiting the generation of monoamine neurotransmitters to specific subcellular locations may also exist in mammals.

## Methods

### Fly stocks and cultivation

The *Hdc^P217^*, *poe^c07064^*, *M(vas-int.Dm) ZH-2A; M(3xP3-RFP.attP) ZH-86Fb* and TRIP RNAi lines for *NSF1* (*P(TRiP.HMS01261)attP2 and P(TRiP.JF01459)attP2*), *unc-104* (*P(TRiP.HMC03512)attP40*), *Khc* (*P(TRiP.GL00330)attP2*), *atg7* (*P(TRiP.HMS01358)attP40*), *atg8a* (*P(TRiP. HMS01328)attP40*), *pros-α5* (*P(TRiP.HMS00119) attP2*), *pros-β1* (*P(TRiP.HMS00139)attP2*), and *poe* (*P(TRiP. HMS00739)attP2*), flies were provided by the Bloomington Drosophila Stock Center. The TRIP transgenic RNAi collection for E3 ubiquitin ligases was obtained from the TsingHua Fly Center and the Bloomington Drosophila Stock Center. The *w^1118^, lovit^1^, nos-Cas9, FRT40A, ey-flp;GMF-hid CL FRT40A/Cyo, GMR-gal4, repo-Gal4, UAS-Hdc-GFP*, and *UAS-tadr* flies were maintained in the lab of Dr. T. Wang at the National Institute of Biological Sciences, Beijing, China (Han et al, 2022). Flies were maintained in 12-h light/12-h dark cycles with ~2000 lux illumination at 25 °C.

### Generation of plasmid constructs and transgenic flies

The *ptrp-attB* vector was generated as described (Bischof et al, 2007; Li and Montell, 2000). To generate the *ptrp-Hdc-mCherry* plasmid, *Hdc* cDNA with a C-terminal mCherry tag was cloned into the *ptrp-attB* vector. To generate *ptrp-emc5-Hdc-mCherry* and

*ptrp-Syt1-Hdc-mCherry* plasmids, the coding regions of fly *emc5* and *Syt1* were amplified from RE09053 and GH01240, respectively, and added to the 5′ end of *ptrp-Hdc-mCherry*. The *hHdc* sequences were synthesized from GENEWIZ, China, and cloned into the *ptrp-attB* vector with a C-terminal mCherry tag to generate the *ptrp-hHdc-mCherry* plasmid. The coding sequences of *emc5* and *Syt1* were added to the 5' end of the *ptrp-hHdc-mCherry* plasmid to generate *ptrp-emc5-hHdc-mCherry* and *ptrp-Syt1-hHdc-mCherry* plasmids. *NSF1* cDNA was amplified from the RE33604 cDNA clone obtained from the DGRC (Drosophila Genomics Resource Center, Bloomington, IN) and cloned into the *pculd* vector with a GFP tag (Desplan et al, 2015). To construct *Syt1-HA-GFP* and *Syt1-strep-GFP* plasmid, the Syt1 cDNA with C-terminal HA tag or Strep-tag was cloned into the *pUAS-GFP-attB* vector. To construct *UAS-HNMT* plasmids, the cDNA of *HNMT* was amplified and subcloned into the *UAST-attB* vector. These constructs were injected into *M(vas-int.Dm) ZH-2A; M(3xP3-RFP.attP) ZH-86Fb* or *M(vas-int.Dm) ZH-2A; M(3xP3-RFP.attP) ZH-51c* embryos, and transformants were identified on the basis of eye color. The *3xP3-RFP and w+* markers were removed by crossing with P(Cre) flies.

To express Hdc and NSF1 in cultured cells, *Hdc* cDNA with a C-terminal mCherry tag and *NSF1* cDNA with a C-terminal myc tag were subcloned into the pIB vector (Invitrogen, Carlsbad, CA) to express in S2 cells.

### Synaptic vesicle purification

The purification of synaptosomes were performed as described with modification (Chantranupong et al, 2020; Depner et al, 2014). Briefly, fly heads were collected and homogenized in homogenization buffer (320 mM sucrose, 4 mM HEPES, pH 7.4) supplemented with protease inhibitor and PMSF. Homogenized heads were centrifuged at $1000 \times g$ for 10 min. The supernatant was then transferred to a new tube and centrifuged at $15,000 \times g$ for 15 min to pellet synaptosomes. The pellet was resuspended in 0.1 ml of homogenization buffer with care to minimize the formation of air bubbles. About 0.9 ml of ice-cold ddH₂O was added, and the liquid was immediately homogenized to lyse synaptosomes. Osmolarity was restored following homogenization by the addition of 5 µl 1 M HEPES. Finally, the homogenate was centrifuged at $17,000 \times g$ for 15 min to remove unbroken synaptosomes and debris. To immuno-isolate synaptic vesicles, 50 µl of suspended, prewashed magnetic HA beads (Thermo Fisher Scientific) were added to the supernatant and incubated at 4 °C with end-over-end rotation for 15 min. For washes, beads were separated with a magnet until the supernatant was clear. Discard the supernatant, and four washes were performed in succession by the addition of 1 ml of KPBS (136 mM KCl, 10 mM KH₂PO4, pH 7.25). The bound proteins were analyzed by western blotting after elution. The following antibody were used at the indicated dilutions: rat anti-Hdc (1:1000) (Han et al, 2022), rabbit anti-GFP (1:2000, EASYBIO, China), rat anti-Tom20 (1:1000)(Zhuang et al, 2016), mouse anti-CSP (1:1000, DSHB), mouse anti-cnx99 (1:1000, DSHB), rabbit anti-Tan (1:1000) (Wagner et al, 2007). Following overnight incubation, the blots were probed with secondary antibodies with IRDye 800 IgG and IRDye 680 IgG (1:5000, LI-COR Biosciences) for 1 h at room temperature. Membranes were then washed before being visualized with an Odyssey infrared imaging system (LI-COR Biosciences).

## Western blotting

Fly heads were homogenized in SDS sample buffer with a pellet pestle (Kimble Chase, Vineland, NJ, USA), and the proteins were fractionated by SDS–polyacrylamide gel electrophoresis. Proteins from the gels were then transferred onto Immobilon-FL transfer membranes (Millipore, Danvers, MA, USA) in a tris-glycine buffer. Membranes were then blocked with 5% milk prepared in TBST (Tris-buffered Saline with Tween 20) and incubated with primary antibodies in TBST overnight at 4 °C with end-over-end rotation. The rat anti-Hdc (1:1000), anti-GFP (1:2000, EASYBIO, China), and mouse anti-actin (1:2000, ABclonal, China) were used as primary antibodies. The blots were subsequently probed with IRDye 800 IgG and IRDye 680 IgG (LI-COR Biosciences, USA). Signals were detected with an Odyssey infrared imaging system (LI-COR Biosciences, USA).

## ERG recordings

ERGs were recorded as described (Desplan et al, 2015). Briefly, two glass microelectrodes were filled with Ringer's solution, and placed on the surfaces of the compound eye and thorax, respectively. The light intensity was ~0.3 mW/cm$^2$, and the wavelength was ~550 nm (source light was filtered using an FSR-OG550 filter, Newport, USA). The electoral signals were amplified with a Warner electrometer IE-210, and recorded with a MacLab/4 s A/D converter and Clampex 10.2 program (Warner Instruments, USA). Fly were dark-adapted for 1 min before recording, and all recordings were carried out at 25 °C.

## Immunohistochemistry

Fly heads were fixed with 4% paraformaldehyde for 2 h at 4 °C or 4% EDAC (for histamine staining), and immersed in 12% sucrose solution overnight at 4 °C as described (Han et al, 2022). The sections (10-μm thick) were prepared from adults that were embedded in an OCT compound (Tissue-Tek, Torrance, CA). Immunolabeling was performed on cryosections sections with rat anti-RFP (1:200, Chromotek, Germany), rat anti-Hdc (1:50) (Han et al, 2022), rabbit anti-LOVIT (1:100) (Xu and Wang, 2019), or anti-CSP (1:100, DSHB), rabbit anti-GFP (1:200, Invitrogen, Carlsbad, CA), and rabbit anti-Tan (1:100) (Wagner et al, 2007) as primary antibodies. For histamine immunolabeling, the rabbit anti-histamine (1:100, ImmunoStar, Hudson, WI) antibody was pre-adsorbed with carcinine, as previously reported (Desplan et al, 2015). Goat anti-rabbit IgG conjugated to Alexa 488 (1:500, Thermo Fisher Scientific, USA), goat anti-mouse IgG conjugated to Alexa 488 (1:500, Thermo Fisher Scientific, USA), goat anti-rat IgG conjugated to Alexa 568 (1:500, Thermo Fisher Scientific, USA), and goat anti-rabbit IgG conjugated to Alexa 647 (1:500, Thermo Fisher Scientific, USA, CA) were used as secondary antibodies. The images were recorded with a Zeiss 800 confocal microscope (Carl Zeiss, Germany).

## Co-immunoprecipitation

Approximately 500 flies expressing mCherry-tagged Hdc or mCherry were collected within 5 days of eclosion. The flies were frozen in liquid nitrogen to dissociate the head from the body, and heads were then sieved out and lysed using lysis buffer (50 mM Tris-HCl, pH 7.5, 150 mM NaCl, 1% NP-40, 1X proteasome inhibitor) at 4 °C. The lysates were then incubated with RFP agarose beads (Chromotek, Germany) for 2 h at 4 °C. After several washes, the beads were incubated with SDS loading buffer, and the supernatant was collected. Sliver staining (Thermo Fisher Scientific, USA) was performed, and bands on the gel were cut out for mass spectra analysis to identify binding proteins.

S2 cells expressing truncated Hdc with mCherry tag and Myc-tagged NSF were collected and lysed with 10 mM Tris-HCl lysis buffer (pH 7.4, 150 mM NaCl, 0.5 mM EDTA, 0.5% NP-40 with 1X proteinase inhibitor) for 30 min. The supernatant was incubated with anti-Myc or RFP agarose beads (Chromotek, Germany), and boiled in SDS loading buffer for further western blot assays. The blots were probed with primary antibodies against Myc (rabbit, 1:2000; Santa Cruz) or RFP (rat 1:1000, Chromotek, Germany), followed by incubation with IRDye 680 goat anti-rabbit IgG (1:10,000, LI-COR Biosciences, USA) and IRDye 800 goat anti-rat IgG (1:10,000, LI-COR Biosciences, USA). Signals were detected using an Odyssey infrared imaging system (LI-COR Biosciences, USA).

## Liquid chromatography–mass spectrometry (LC-MS)

To quantify the total histamine content in the head, fly heads were collected and subjected to LC-MS experiments as previously described (Han et al, 2017). Briefly, the Dionex Ultimate 3000 UPLC system was coupled to a Quantiva ultra triple-quadrupole mass spectrometer (Thermo Fisher, USA), equipped with a heated electrospray ionization (HESI) probe in negative ion mode. Extracts were separated by a 2.5-μm Fusion-RP C18 column (Phenomenex, USA). Data were acquired in selected reaction monitoring for histamine with transitions of 112/95.2. The source parameters are as follows: spray voltage, 3000 V; ion transfer tube temperature, 350 °C; vaporizer temperature, 300 °C; sheath gas flow rate, 40 Arb; auxiliary gas flow rate, 20 Arb; CID gas, 2.0 mTorr. Data analysis and quantification were performed using the Xcalibur 3.0.63 software (Thermo Fisher, USA). Each sample contained ~50 *Drosophila* heads, and the mean values from three samples were calculated.

## The phototaxis assay

Flies were dark-adapted for 15 min before the phototaxis assay as previously described (Desplan et al, 2015). Briefly, a transparent glass tube 20 cm in length and 2.5 cm in diameter was used, and a light source (with a light intensity of ~6000 lux) was placed at one end of the glass tube. The tube was placed horizontally in the dark, and flies were gently tapped into the no-light end of the tube. The light was turned on, and the number of flies that walked past a 10-cm mark on the tube within 30 s was counted. The phototaxis index was calculated by dividing the number of flies that walked past the mark by the total number of flies. Five groups of flies were collected for each genotype, and three repeats were made for each group. Each group consisted of at least 20 flies, and results were expressed as the mean for the five groups.

## RNA extraction and qPCR

Fly head, retina, and lamina were dissected, and total RNA was prepared from the dissected tissues using Trizol reagent (Thermo Fisher Scientific, USA). Total cDNA was synthesized using a cDNA

Synthesis kit (TransGen Biotech), and real-time PCR was performed using iQ SYBR green supermix (Bio-Rad Laboratories, USA). Three different samples were collected from each genotype. Quantifications of transcripts were normalized to rp49. The following primers were used:

*mCherry*: forward, 5′- CCTGTCCCCTCAGTTCATGT; reverse, 5′-CCCAT GGTCTTCTTCTGCAT; *rp49*: forward, 5′-ACAGGCCCAA-GATCGTGAAG; reverse, 5′-CTTGC GCTTCTTGGAGGAGA.

## Quantification and statistical analysis

All experiments were repeated as indicated in each figure legend ($n \geq 3$). All statistical analyses were performed using GraphPad Prism 8. The variations of data were evaluated as mean ± SD. The statistical significance of the differences between the two groups was measured by the unpaired two-tailed Student's *t*-test. Dunnett's one-way ANOVA was performed for multiple comparisons against a control group. Tukey's one-way ANOVA was used for multiple comparisons with each other. The value of $p < 0.05$ was considered statistically significant (ns, not significant; $*p < 0.05$; $**p < 0.01$; $***p < 0.001$; $****p < 0.0001$).

## Data availability

The source data of micro images in this paper has been submitted to BioImages Archive. The accession number is S-BIAD1238.

The source data of this paper are collected in the following database record: biostudies:S-SCDT-10_1038-S44318-024-00223-0.

## Peer review information

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

## Acknowledgements

We thank the Bloomington Stock Center, the Tsinghua Fly Center, and the Developmental Studies Hybridoma Bank for stocks and reagents. We thank Y. Wang and X. Liu for assistance with fly injections, the Metabolomics Center at the Institute of Biological Sciences for lipidomic mass spectrometric analysis, the Electron Microscopy Center at the Institute of Biological Sciences, and Dr. Lin Yang from the Institute of Genetics and Developmental Biology for assistance with TEM, and the Proteomics Facility at the Institute of Biological Sciences for Mass Spectrometry support. We thank Dr. D. O'Keefe for editing the manuscript. This work was supported by grants from the National Natural Science Foundation of China (81870693), institution grants from the Beijing Municipal Commission of Science and Technology Commission, and the Chinese Ministry of Science and Technology awarded to T. Wang.

## Author contributions

**Lei Peng**: Conceptualization; Investigation; Visualization; Methodology; Writing—original draft; Writing—review and editing. **Tao Wang**: Conceptualization; Supervision; Writing—original draft; Writing—review and editing.

Source data underlying figure panels in this paper may have individual authorship assigned. Where available, figure panel/source data authorship is listed in the following database record: biostudies:S-SCDT-10_1038-S44318-024-00223-0.

## Disclosure and competing interests statement

The authors declare no competing interests.

# Expanded View Figures

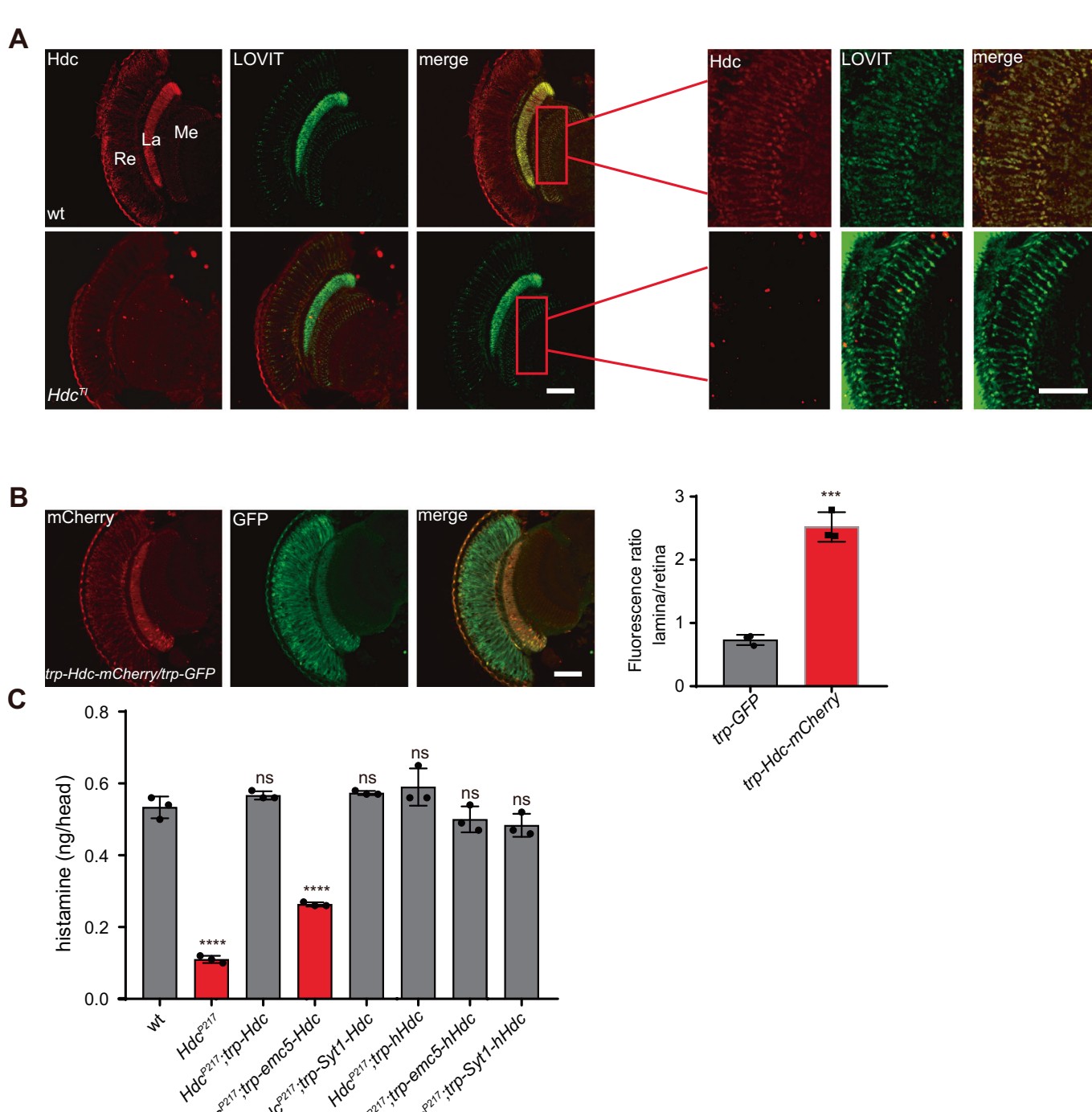

**Figure EV1.  Expression of Hdc in both soma and axon restores total histamine content.**

(A) Head longitudinal sections of wild-type (wt) and Hdc null mutants (*Hdc^TI*) were labeled with Hdc and LOVIT antibodies. Sections of the distal medulla neuropil are shown on the right. Scale bars, 20 μm on the left and 50 μm on the right. La lamina, Me medulla, Re retina. (B) Head longitudinal sections from flies expressing both Hdc-mCherry and GFP in photoreceptor cells under the control of the *trp* promoter (*trp-Hdc-mCherry/trp-GFP*). Sections were labeled with antibodies against mCherry and GFP. *trp-GFP* vs *trp-Hdc-mCherry* ***p = 0.0002. Unpaired *t*-test, n = 3, mean ± sd. Scale bars, 20 μm. (C) Total histamine levels in heads of wild-type, Hdc mutant flies (*Hdc^P217^*), and *Hdc^P217^* flies expressing mCherry-tagged Hdc (*Hdc^P217^;trp-Hdc*), EMC5-Hdc (*Hdc^P217^;trp-emc5-Hdc*), Syt1-Hdc (*Hdc^P217^;trp-Syt1-Hdc*), hHdc (*Hdc^P217^;trp-hHdc*), EMC5-hHdc (*Hdc^P217^;trp-emc5-hHdc*), or Syt1-hHdc (*Hdc^P217^;trp-Syt1-hHdc*) driven by the *trp* promoter. Each sample contained 20 fly heads and the mean values from three samples were calculated. wt vs *Hdc^P217^* ****p < 0.0001, wt vs *Hdc^P217^;trp-Hdc* p = 0.5571, wt vs *Hdc^P217^;trp-emc5-Hdc* ****p < 0.0001, wt vs *Hdc^P217^;trp-Syt1-Hdc* p = 0.3802, wt vs *Hdc^P217^;trp-hHdc* p = 0.1151, wt vs *Hdc^P217^;trp-emc5-hHdc* p = 0.5571, wt vs *Hdc^P217^;trp-Syt1-hHdc* p = 0.1917. Dunnett's one-way ANOVA, mean ± sd, ns not significant.

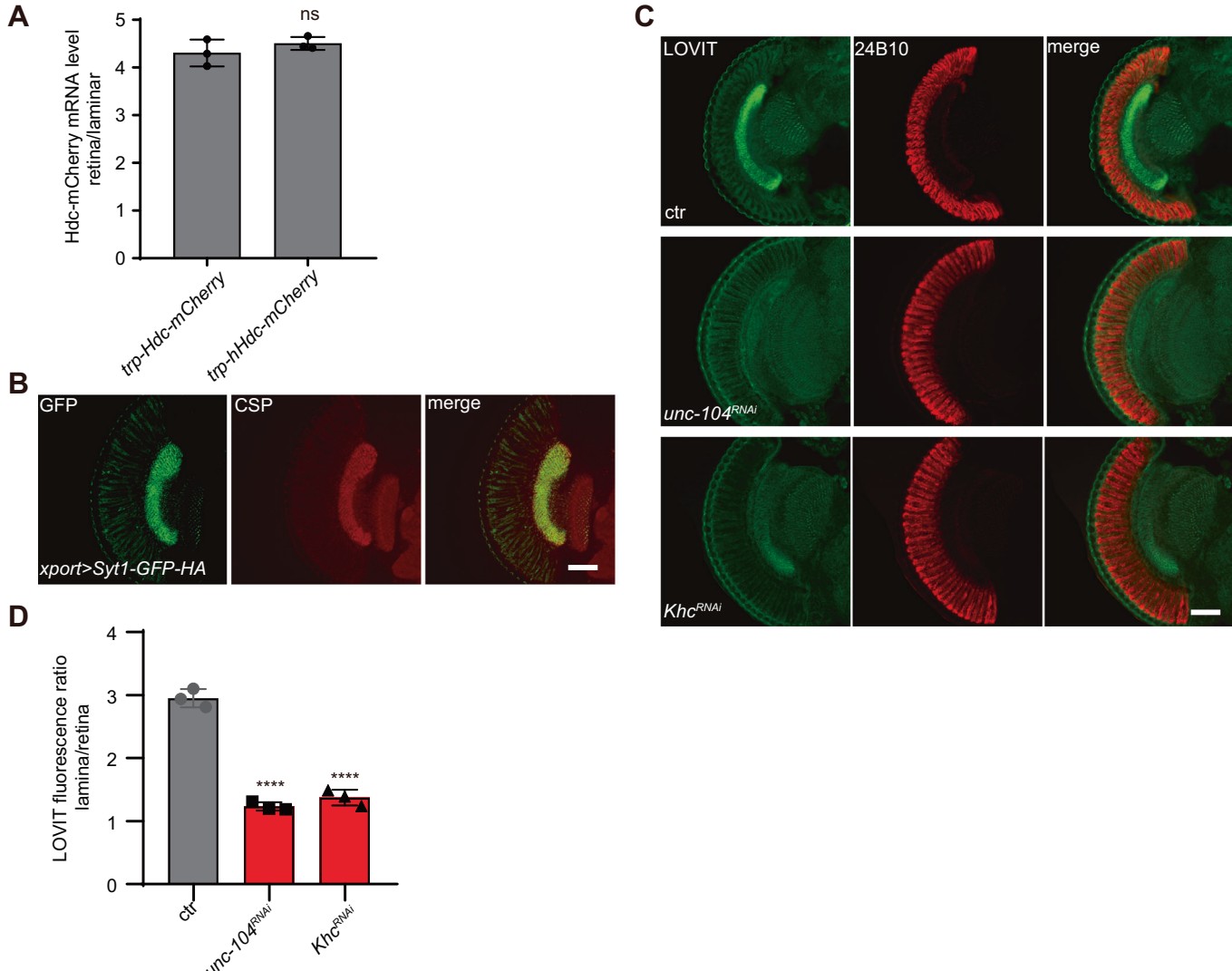

**Figure EV2. Disruption of the kinesin motor reduces axonal synaptic vesicles.**

(A) Hdc-mCherry mRNA level ratio between the lamina and retina of *trp-Hdc-mCherry* and *trp-hHdc-mCherry* flies. $p = 0.3287$. Unpaired *t*-test, $n = 3$, mean ± sd, ns not significant. (B) Immunostaining of the head section of *xport>Syt1-GFP-HA* flies (*xport-Gal4/UAS-Syt1-GFP-HA*) showed Syt1-GFP-HA colocalized with the synaptic vesicle marker, CSP. Scale bars, 20 μm. (C) Head cross-sections of control (*GMR>GFP^RNAi*, *GMR-Gal4/UAS-GFP^RNAi*), *GMR>unc-104^RNAi* (*GMR-Gal4/UAS-unc-104^RNAi*), and *GMR>Khc^RNAi* (*GMR-Gal4/UAS-Khc^RNAi*) were labeled with LOVIT antibodies (synaptic vesicles marker, green) and 24B10 (photoreceptor marker, red). Scale bars, 20 μm. (D) Quantification of LOVIT signals in lamina versus retina. Three fly head sections were used for quantification. ctr vs *unc-104^RNAi* ****$p < 0.0001$, ctr vs *Khc^RNAi* ****$p < 0.0001$. Dunnett's one-way ANOVA, mean ± sd.

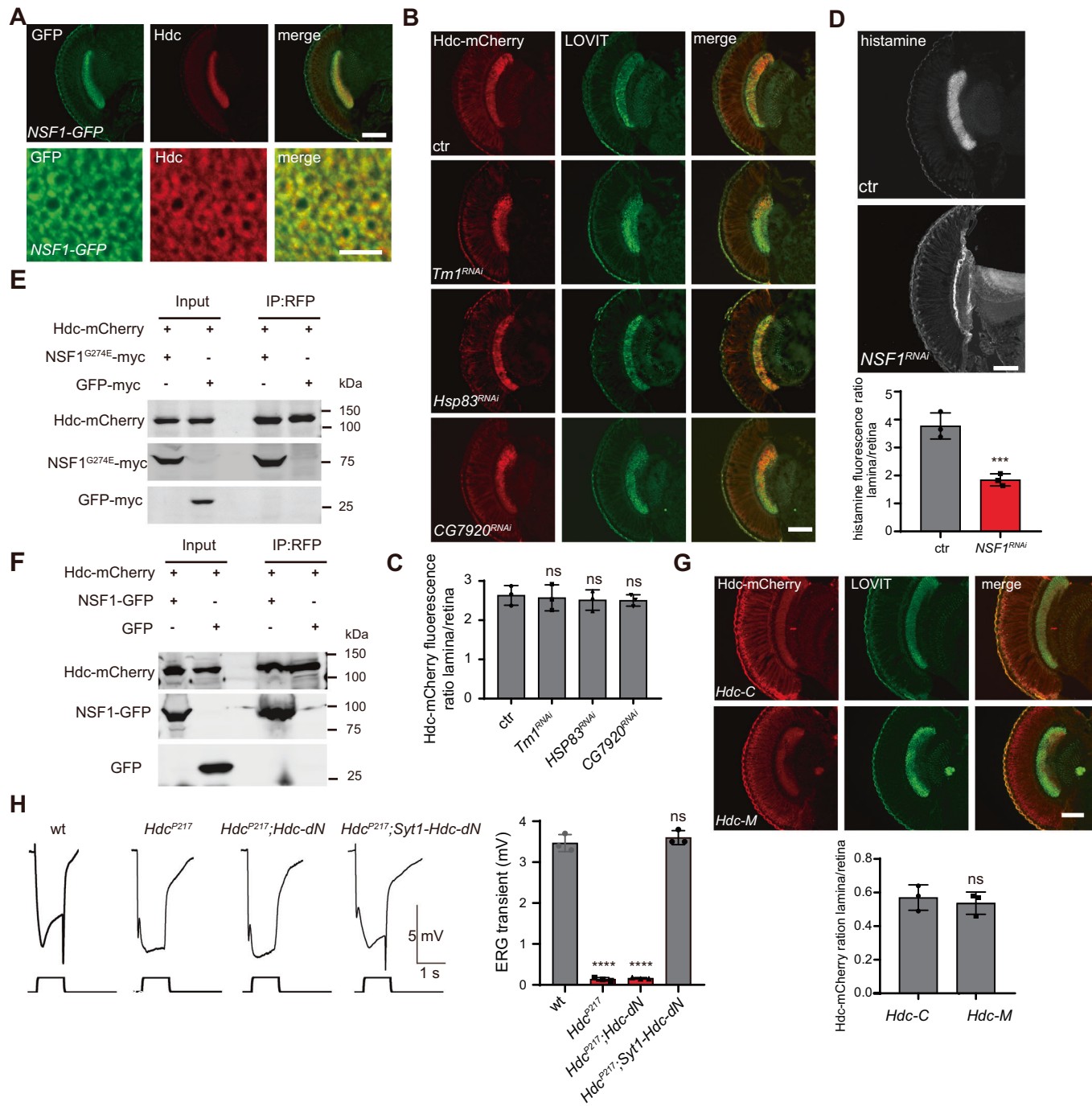

**Figure EV3. Interaction between NSF1 and Hdc through its N-terminals is required for localization and function of Hdc at synaptic terminals.**

(A) Head horizontal sections of flies expressing GFP-tagged NSF1 under a weak photoreceptor cell promoter (*culd-NSF1-GFP*) were labeled for GFP and Hdc. Cross-sections of the lamina are shown on the bottom. Scale bars, 20 μm at the top and 50 μm at the bottom. (B) Head sections of *trp-Hdc-mCherry* flies expressing *GFP^RNAi* (*GMR-Gal4/UAS-GFP^RNAi*), *Tm1^RNAi* (*GMR-Gal4/UAS-Tm1^RNAi*), *Hsp83^RNAi* (*GMR-Gal4/UAS-Hsp83^RNAi*), or *CG7920^RNAi* (*GMR-Gal4/UAS-CG7920^RNAi*) were stained with mCherry antibodies. Scale bar, 20 μm. (C) Quantitation of Hdc-mCherry fluorescence ratio between lamina and retina. ctr vs *Tm1^RNAi* $p = 0.9871$, ctr vs *Hsp83^RNAi* $p = 0.8907$, ctr vs *CG7920^RNAi* $p = 0.8675$. Dunnett's one-way ANOVA, $n = 3$, mean ± sd, ns not significant. (D) Horizontal sections of wild-type (wt) and flies expressing *NSF1^RNAi* (*NSF1^RNAi*) were labeled using antibodies against histamine. Scale bars, 20 μm. Quantification of fluorescence intensity ratios of histamine between the lamina and retina was shown on the right. ctr vs *NSF1^RNAi* ***$p = 0.0005$. Unpaired t-test, $n = 3$, mean ± sd. (E) Interaction of Hdc and NSF1 is independent of the ATPase activity of NSF1. Hdc-mCherry was co-expressed with NSF1^G274E-Myc or GFP-Myc in S2 cells. Cell lysates were immunoprecipitated with anti-mCherry beads, and blotted against either mCherry or Myc. (F) Hdc interacts with NSF1 in vivo. Cell lysates from dissected heads of *trp-Hdc-mCherry/culd-NSF1-GFP* flies or *trp-Hdc-mCherry/trp-GFP* flies were immunoprecipitated with anti-mCherry beads, and blotted against mCherry and GFP. (G) Immunostaining of head sections of flies expressing mCherry-tagged truncated Hdc-M (*trp-Hdc-M-mCherry*) and Hdc-C (*trp-Hdc-C-mCherry*) showed a reduction of mCherry signals in lamina. Hdc-C vs Hdc-M $p = 0.5970$. Unpaired t-test, $n = 3$, mean ± sd, ns not significant. (H) ERG recorded from *Hdc^P217* flies expressing Hdc-dN (*Hdc^P217;trp-Hdc-dN*) and Syt1-Hdc-dN (*Hdc^P217;trp-Syt1-Hdc-dN*). OFF transients were quantified based on data from three flies. wt vs *Hdc^P217* ****$p < 0.0001$, wt vs *Hdc^P217;trp-Hdc-dN* ****$p < 0.0001$, wt vs *Hdc^P217;trp-Syt1-Hdc-dN* $p = 0.5272$. Dunnett's one-way ANOVA, mean ± sd, ns not significant.

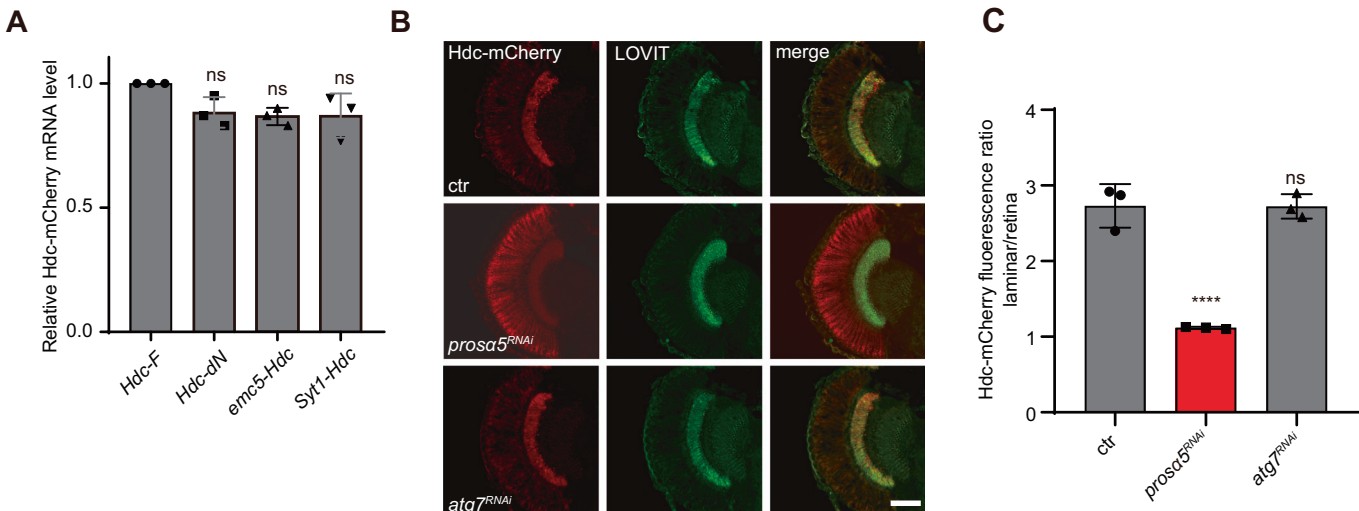

**Figure EV4.  Disruption of the proteasome but not lysosomes increases Hdc levels in the retina.**

(**A**) QPCR comparison of mRNA levels of truncated Hdc. Total RNA was extracted from heads of Hdc-F (*trp-Hdc-mCherry*), Hdc-dN (*trp-Hdc-dN-mCherry*), emc5-Hdc (*trp-emc5-Hdc-mCherry*), and Syt1-Hdc (*trp-Syt1-Hdc-mCherry*) flies. A primer pair against *mCherry* sequences was used for QPCR, and results were normalized to *rp49*. Hdc-F vs Hdc-dN $p = 0.0854$, Hdc-F vs emc5-Hdc $p = 0.0504$, Hdc-F vs Syt1-Hdc $p = 0.0557$. Dunnett's one-way ANOVA, $n = 3$, mean ± sd, ns not significant. (**B**) Head horizontal sections of *trp-Hdc-mCherry* flies expressing *GFP*[RNAi] (*GMR>GFP*[RNAi], *trp-Hdc-mCherry GMR-Gal4/UAS-GFP*[RNAi]), *prosa5*[RNAi] (*GMR>prosa5*[RNAi], *trp-Hdc-mCherry GMR-Gal4/UAS-prosa5*[RNAi]), and *atg7*[RNAi] (*GMR>atg7*[RNAi], *trp-Hdc-mCherry GMR-Gal4/UAS-atg7*[RNAi]) were labeled with mCherry (red) and LOVIT (green) antibodies. Scale bars, 20 μm. (**C**) Quantification of Hdc-mCherry fluorescence ratio between lamina and retina. Three fly head sections were used for quantification. ctr vs *prosa5*[RNAi] ****$p < 0.0001$, ctr vs *atg7*[RNAi] $p = 0.9986$. Dunnett's one-way ANOVA, mean ± sd, ns not significant.

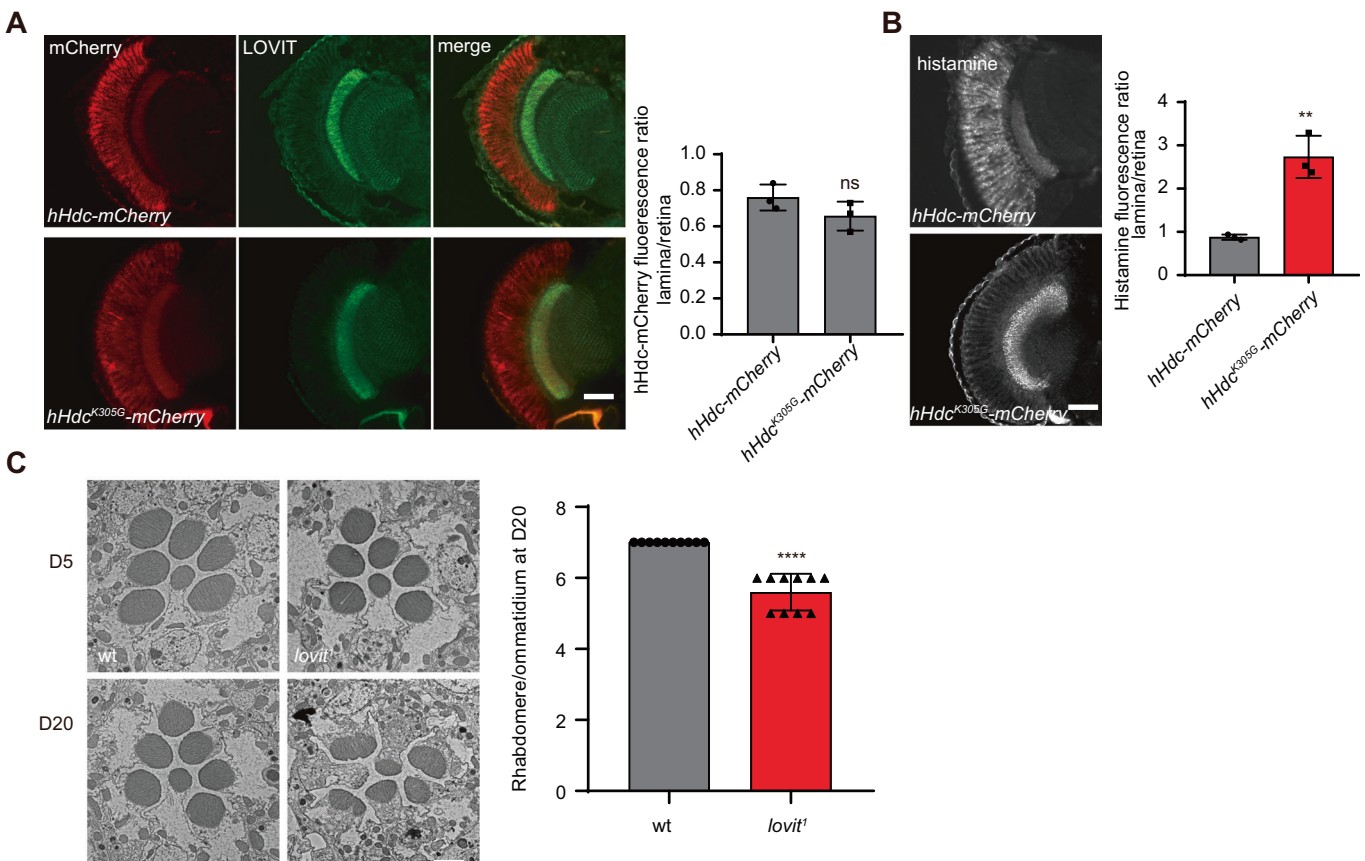

**Figure EV5.** **The Hdc^K305G mutant cannot synthesize histamine.**

(A) Immunostaining of mCherry in *trp-hHdc-mCherry* and *trp-hHdc^K305G-mCherry* flies revealed no changes in the distribution and levels of ectopic hHdc proteins. *hHdc-mCherry* vs *hHdc^K305G-mCherry* p = 0.1738. Unpaired *t*-test, n = 3, mean ± sd. (B) Histamine was immunolabeled in horizontal sections of heads of *trp-hHdc-mCherry* and *trp-hHdc^K305G-mCherry* flies. Quantification of histamine fluorescence intensity ratios between the entire lamina and retina were based on three sections. *hHdc-mCherry* vs *hHdc^K305G-mCherry* **p = 0.0028. Unpaired *t*-test, mean ± sd. (C) TEM images from wildtype and *lovit^1* flies that were 1 (upper) or 20 days old (bottom). Scale bar, 2 μm. Ten sections were used for quantification. wt vs *lovit^1* ****p < 0.0001. Unpaired *t*-test, mean ± sd.

