## [Peer Review File · The EMBO Journal]

Histamine Synthesis and Transport are Coupled in Axon Terminals via Double Quality Control System.

Lei Peng and Tao Wang

Corresponding author: Tao Wang (wangtao1006@nibs.ac.cn)

Review Timeline:

Submission Date:	5th Mar 24
Editorial Decision:	29th Apr 24
Revision Received:	20th Jun 24
Editorial Decision:	22nd Jul 24
Revision Received:	26th Jul 24
Accepted:	7th Aug 24

Editor: Kelly Anderson

Transaction Report:

Dear Prof. Wang,

Thank you for submitting your manuscript for consideration by the EMBO Journal. It has now been seen by three referees whose comments are shown below.

Given the referees' positive recommendations, I would like to invite you to submit a revised version of the manuscript, addressing the comments of all three reviewers. I should add that it is EMBO Journal policy to allow only a single round of revision, and acceptance of your manuscript will therefore depend on the completeness of your responses in this revised version. It would be good to discuss your plan to address the referee concerns and I am available to do so in the coming weeks by zoom or email.

Thank you for the opportunity to consider your work for publication. I look forward to your revision.

Yours sincerely,

Kelly M Anderson, PhD
Editor, The EMBO Journal
k.anderson@embojournal.org

We realize that it is difficult to revise to a specific deadline. In the interest of protecting the conceptual advance provided by the work, we recommend a revision within 3 months (28th Jul 2024). Please discuss the revision progress ahead of this time with the editor if you require more time to complete the revisions.

Referee #1:

The manuscript by Peng and Wang addresses the issue of monoamine neurotransmitter toxicity which has attracted considerable attention especially in the context of dopamine and its role in Parkinson's disease. In their manuscript the authors use the *Drosophila* photoreceptors and their major neurotransmitter histamine to address this issue. Specifically, the authors aim to show that the tight coupling of HDC, the enzyme that generates histamine in photoreceptors, requires tight coupling to synaptic vesicles to prevent histamine toxicity. Using different genetic approaches, the authors demonstrate that elevated histidine decarboxylase activity in the cell bodies of photoreceptors is toxic, whether this is due to the expression of "untethered" *Drosophila* or human HDC proteins or due to their lack of degradation in mutations of the POE ubiquitin E3 ligase. Furthermore, the authors convincingly demonstrate that toxicity depends on elevated levels of the substrate histamine and not elevated levels of the enzyme itself. This is an exciting study and I have no major concerns that could interfere with its publication.

There are some minor issues that need to be addressed:

- Figure 1 C as displayed is very confusing. It is not clear how the ratio between lamina and retina level of histamine can yield a negative number. Maybe just show the red and grey bars next to each other?
- Figure 2, the panel 2A appears to have be mislabeled, as it indicates that anti-HA IP pulled down synaptic vesicles tagged with Syt-GFP-strep rather than Syt-GFP-HA.
- In Table S2, that summarizes the ubiquitin ligase screen, it is not specified what "Results: YES/NO" exactly means. Presumably, it refers to the level of HDC by Western blot analysis. If so, one might expect that some of the ligases knock-downs exhibited lower levels of HDC and, if so, that should be mentioned.
- In the same table, it is not clear from where the authors have derived the information about the expression levels in the eye for different ubiquitin ligases.
- the legend to Figure S6 has a mistake. The authors write: "Carcinine is imported into presynaptic terminals by TADR and converted to histamine by Hdc localized to synaptic vesicles." In fact, it is histidine, not carcinine, that is imported by TADR, as the authors correctly mention in other parts of the manuscript.
- Also, why not include this Model Figure (S6) in the main manuscript instead in the supplement? I think most readers would benefit.
- a few typos that should be fixed (what be made easier with page numbers!)
 1. Methods section "Liquid chromatography-mass spectrometry (LC-MS)" starts with "To measurement" instead of "to measure "
 2. "In photoreceptor neurons, Hdc does not physically interacts with LOVIT," should be
- does not interact -
 3. "By contrast, GFP expressed under via the trp promoter localized to the retina,"
- under control off?

Referee #2:

This is a very interesting study which shows that the localization of histidine decarboxylase (*hdc*) is important for histamine storage, release and vision. This has been proposed a number of times before with other transmitters but the data has never been convincing. The authors here provide compelling evidence that localization of *hdc* to the nerve terminal (and presumably to synaptic vesicles) is required, and involves both axonal targeting and the degradation of mistargeted enzyme in the cell body. The combination of immunostaining for the decarboxylase, histamine, ERG and behavior demonstrates the robustness of the findings and importance of this mechanism. Use of the human enzyme to bypass the degradation of mistargeted *hdc* is an elegant way to corroborate these findings. The control for lack of effect on synaptic vesicle localization is also appropriate. In addition, the authors provide strong evidence of a role for the proteasome (not the lysosome or at least autophagy) and a

specific ubiquitin ligase in the quality control required to remove mislocalized hdc. This is a very nice body of work.

On the other hand, the evidence of a role for NSF in axonal targeting of hdc is not clear. The authors show effects of two NSF knockdowns on axonal hdc and apparently little effect on synaptic vesicle distribution. However, NSF should affect vesicle recycling and other presynaptic processes, so specificity would be surprising. Indeed, the control LOVIT staining does seem impaired with the second knockdown (Fig. 3B). The biochemistry is also unclear. In Fig. 3A, a much better control would be flies without Hdc-mCh-of course, NSF-myc can only be pulled down in cells expressing it and it would be much better to test dependence on the protein used for pull down. The N-terminal deletion could provide some evidence of specificity but Fig. 3C also has problems because the N-terminal deletion of hdc is smaller than full length and would likely have migrated out of the slice of gel show in the IP. There is also no demonstration of effects of NSF on histamine staining, ERG or behavior. It is also likely that the effect of NSF is indirect. NSF is a soluble protein and although a fraction associates with synaptic vesicles, it seems likely that its AAA-ATPase activity is required to maintain the folding or function of other components that target hdc to vesicles, rather than serving itself as the interacting protein-hence the importance of controls for the immunoprecipitation.

Additional concerns:

Why does hHdc restore axonal histamine but not the ERG? Seems like partial rescue since phototaxis was partially restored. The panels to the right in Fig. 2A seem to be reversed-the strep-tagged syt1 seems to be pulled down by HA whereas the HA-tagged is not.

There are also a number of misstatements in the introduction and elsewhere:

Dense core vesicles are actually most abundant in axons, although not always presynaptic.

All of Tourettes is not caused by a mutation in Hdc.

DOPA does not accelerate PD-it is just associated with side effects of treatment.

Referee #3:

Summary:

In this paper, Peng and Wang investigate the mechanisms of histamine synthesis and packaging into synaptic vesicles in *Drosophila* photoreceptors and the restriction of this process to axon terminals.

The authors showed the importance of local synthesis of histamine and its role as an important contributor to the synthesis of Hdc in the axon terminals for a normal synaptic transmission, hence the processing of visual input. They then explored their hypotheses on the localization of Hdc in synaptic terminals and discovered that the mature proteins were transported to the axon terminals via synaptic vesicles by proving their physical interaction. Furthermore, they identified NSF1 as the protein to which Hdc binds for its transport via synaptic vesicles. Additionally, they explored the quality control mechanisms for correct Hdc localization. After confirming that mislocalized Hdc is degraded through the ubiquitin-proteasome system, they performed a gene screening for E3 ubiquitin ligases, one of the key factors in the ubiquitin-proteasome system, and found that Purity of essence is required for Hdc degradation. Finally, they showed that knocking down these quality control mechanisms leads to the mislocalization of Hdc and hence histamine, which can cause retinal degradation, together illustrating the significance of the process of transporting Hdc to axon terminals.

This study, through a well-structured, wide variety of experiments, advances our understanding of histamine synthesis and localization mechanisms within *Drosophila* photoreceptors, shedding light on fundamental processes underlying synaptic transmission and emphasizing the crucial role of local histamine synthesis in visual processing.

Major Comments:

1) The experiments with Hdc mutant fly lines lack explicit reference to the mutation throughout the main text, and the implications of this mutation on protein function remain unexplained. While images indicate alterations in Hdc levels and histamine synthesis, further clarification on the effects of the mutation and a reference is needed.

2) The NSF1 and Hdc coimmunoprecipitation experiments were done in cell culture and are well explained. Moreover, they confirm the NSF knock-down experiments. However, although the fly lines are available and used in the experiments, an explanation is lacking why it would not be feasible to do the experiments in flies.

Minor Comments:

1) Results, second paragraph: 'By contrast, GFP expressed under via the trp promoter localized to the retina, lamina, and medulla'. 'via' must go. There are a couple of other typos in the ms as well.

2) A minor issue pertains to the naming of Hdc mutants in figures. Utilizing mutant names, such as HdcP217, rather than solely

'Hdc,' could enhance clarity.

3) In Figure S3, D and E can be rearranged for a cleaner separation.

Non-essential Suggestion:

1) The conclusory statements in several paragraphs, including the abstract and discussion appear somewhat tentative and not strong enough despite the robust evidence presented.

Conclusion:

In conclusion, Peng and Wang's study advances our understanding of histamine synthesis and localization mechanisms in *Drosophila* photoreceptors, providing valuable insights into their importance in synaptic transmission, thus, visual processing. While the concerns raised are relatively minor, addressing them would further enhance the clarity and impact of their findings, ensuring the article reaches its full potential.

Referee #1:

The manuscript by Peng and Wang addresses the issue of monoamine neurotransmitter toxicity which has attracted considerable attention especially in the context of dopamine and its role in Parkinson's disease. In their manuscript the authors use the *Drosophila* photoreceptors and their major neurotransmitter histamine to address this issue.

Specifically, the authors aim to show that the tight coupling of HDC, the enzyme that generates histamine in photoreceptors, requires tight coupling to synaptic vesicles to prevent histamine toxicity. Using different genetic approaches, the authors demonstrate that elevated histidine decarboxylase activity in the cell bodies of photoreceptors is toxic, whether this is due to the expression of "untethered" *Drosophila* or human HDC proteins or due to their lack of degradation in mutations of the POE ubiquitin E3 ligase.

Furthermore, the authors convincingly demonstrate that toxicity depends on elevated levels of the substrate histamine and not elevated levels of the enzyme itself. This is an exciting study and I have no major concerns that could interfere with its publication.

There are some minor issues that need to be addressed:

- Figure 1 C as displayed is very confusing. It is not clear how the ratio between lamina and retina level of histamine can yield a negative number. Maybe just show the red and grey bars next to each other?

Thank the reviewer for the suggestions in Figure 1C. We replace it with the new Figure 1C.

- Figure 2, the panel 2A appears to have be mislabeled, as it indicates that anti-HA IP pulled down synaptic vesicles tagged with Syt-GFP-strep rather than Syt-GFP-HA. We apologize for the mistake. The mislabeled Figure 2A has been replaced with the correct one, and we go through the manuscript a few times to eliminate these kinds of mistakes.

- In Table S2, that summarizes the ubiquitin ligase screen, it is not specified what "Results: YES/NO" exactly means. Presumably, it refers to the level of HDC by Western blot analysis. If so, one might expect that some of the ligases knock-downs exhibited lower levels of HDC and, if so, that should be mentioned.

- In the same table, it is not clear from where the authors have derived the information about the expression levels in the eye for different ubiquitin ligases.

Thanks for the suggestions. We reasoned that knocking down E3 ligase resulted in Hdc-mCherry accumulation in the cell body. Therefore, we directly examined the Hdc-mCherry fluorescence using a stereo microscope. "YES" indicated increased mCherry signals and no reduced Hdc-mCherry signals were characterized from this screen. The information on tissue expression level was collected from the "Expression Data" section in FlyBase.

We then added a caption to the Table S2 as "Table 2. E3 ligases screen. The levels of Hdc-mCherry were assessed via stereo microscope, with "YES" indicating increased mCherry signals. No instances of reduced Hdc-mCherry signals were observed. Tissue

expression data were sourced from the "Expression Data" section in FlyBase.".

- the legend to Figure S6 has a mistake. The authors write: "Carcinine is imported into presynaptic terminals by TADR and converted to histamine by Hdc localized to synaptic vesicles." In fact, it is histidine, not carcinine, that is imported by TADR, as the authors correctly mention in other parts of the manuscript.

We apologize for the mistake. The incorrect legend for Figure S6 (new Figure 6) has been replaced as "Histidine is imported into presynaptic terminals by TADR, and then converted to histamine by Hdc localized to synaptic vesicles."

- Also, why not include this Model Figure (S6) in the main manuscript instead in the supplement? I think most readers would benefit.

Thanks for this suggestion, and we now moved Figure S6 to the main Figure as Figure 6.

- a few typos that should be fixed (what be made easier with page numbers!)

1. Methods section "Liquid chromatography-mass spectrometry (LC-MS)"

starts with "To measurement" instead of "to measure "

2. "In photoreceptor neurons, Hdc does not physically interacts with LOVIT," should be
- does not interact -

3. "By contrast, GFP expressed under via the trp promoter localized to the retina,"
- under control off?

We apologize for these mistakes. The correct ones have been highlighted in the article. Additionally, we have thoroughly reviewed the article to correct misstatements. Thank you so much for your advice.

Referee #2:

This is a very interesting study which shows that the localization of histidine decarboxylase (hdc) is important for histamine storage, release and vision. This has been proposed a number of times before with other transmitters but the data has never been convincing. The authors here provide compelling evidence that localization of hdc to the nerve terminal (and presumably to synaptic vesicles) is required, and involves both axonal targeting and the degradation of mistargeted enzyme in the cell body. The combination of immunostaining for the decarboxylase, histamine, ERG and behavior demonstrates the robustness of the findings and importance of this mechanism. Use of the human enzyme to bypass the degradation of mistargeted hdc is an elegant way to corroborate these findings. The control for lack of effect on synaptic vesicle localization is also appropriate. In addition, the authors provide strong evidence of a role for the proteasome (not the lysosome or at least autophagy) and a specific ubiquitin ligase in the quality control required to remove mislocalized hdc. This is a very nice body of work.

On the other hand, the evidence of a role for NSF in axonal targeting of hdc is not clear. The authors show effects of two NSF knockdowns on axonal hdc and apparently little effect on synaptic vesicle distribution. However, NSF should affect vesicle recycling and

other presynaptic processes, so specificity would be surprising. Indeed, the control LOVIT staining does seem impaired with the second knockdown (Fig. 3B).

We thank reviews for the concern. Vesicle enrichment in the synaptic terminal is mediated by kinesin-dependent axonal trafficking, and NSF1 is not involved in the trafficking and concentration of synaptic vesicles. NSF1 is indeed involved in SVs fusion to the plasma membrane. Therefore, the loss-of-function mutations of NSF1 result in the accumulation of docked vesicles, while the total number of SVs in synaptic terminals are unaffected (Kawasaki et al. 1998; PMID: 9852561). The second NSF1 knockdown as displayed led to misunderstanding due to the angle of the cross-sections. We now replace it with a new image in new Fig. 3A. In conclusion, as shown in Fig. 3A right panel, the quantification of LOVIT signals between lamina and retina is unchanged in both *NSF1^{RNAi}* lines, which is consistent with the function of NSF1 in the fusion of SVs and new identified Hdc trafficking. However, both Hdc signals and Histamine levels in lamina are significantly reduced supporting the critical role of NSF1 in trafficking of Hdc to synaptic terminals. We added a few sentences of discussion on Page 10 high-lighted.

The biochemistry is also unclear. In Fig. 3A, a much better control would be flies without Hdc-mCh-of course, NSF-myc can only be pulled down in cells expressing it and it would be much better to test dependence on the protein used for pull down.

Thanks for your concern about the co-IP assay. The co-IP assay in Figure 3A (new Figure 3B) was performed in S2 cells. We expressed Hdc-mCherry and NSF-myc in S2 cells and NSF1 can be pulled down with Hdc. The results showed that Hdc directly interacted with NSF1 (new Figure 3B). We also did a co-IP assay in flies. Briefly, we performed the co-IP experiments using fly tissues expressing both NSF1-GFP and Hdc-mCherry (*trp-Hdc-mCherry/culd-NSF1-GFP* flies), and as a result, the Hdc-mCherry efficiently interacted with NSF1-GFP but not with GFP *in vivo*, confirming the interaction of NSF1 and Hdc (reviewer figure below). We generated a new Figure EV3F, and added this result to the results section (Page 7 highlighted).

The N-terminal deletion could provide some evidence of specificity but Fig. 3C also has problems because the N-terminal deletion of *hdc* is smaller than full length and would likely have migrated out of the slice of gel shown in the IP.

Thanks for your concern about the assay in Figure 3C. The N-terminal is 34 amino acids. The deletion of the N terminal is a little bit smaller than the full length. The full-length Hdc-mcherry is about 130 kDa and Hdc-dN (deletion of N terminal) is less than 130 but larger than 100 kDa. We added the marker of 130 and 100 kDa to the Figure 3C. Therefore, the Hdc-dN cannot migrate out of the gel shown in the IP. Please check Figure 3C for details.

There is also no demonstration of effects of NSF on histamine staining, ERG or behavior. It is also likely that the effect of NSF is indirect. NSF is a soluble protein and although a fraction associates with synaptic vesicles, it seems likely that its AAA-ATPase activity is required to maintain the folding or function of other components that target *hdc* to vesicles, rather than serving itself as the interacting protein-hence the importance of controls for the immunoprecipitation.

Thanks for these suggestions, and we performed the experiments as suggested. In summary, in the NSF1 knock-down flies, the histamine level in synapse was significantly decreased. NSF1^{G274E} was a reported NSF1 mutant form and showed no AAA-ATPase activity (Müller, J. M. et al, 1999; PMID: 10559959). We then expressed the NSF1^{G274E} in S2 cells and found that NSF1^{G274E} could still interact with Hdc, which demonstrated that AAA-ATPase activity of NSF1 was not involved in Hdc targeting to vesicles (Reviewer Figure B below). We added these results to the new Figure EV3E, and the result sections on Page 7, highlighted. Please check the new Figure EV3E and text for details.

As the reviewer suggested, we stained the histamine of the NSF1^{RNAi} sections. As results, consistent with the mis-localization of Hdc, histamine failed to concentrate in synaptic terminals of flies expressing NSF1^{RNAi} (Reviewer Figure A below), and we added these results to Figure EV3D and associated text (on Page 7 highlighted). Furthermore, we also checked the ERG responses and phototaxis behaviors of the NSF1^{RNAi} expressing flies, and both ERG responses and phototaxis behaviors are impaired by NSF1^{RNAi} (See the review Figure C and D below). However, NSF1 was well-established to have a key role in SV release, which also caused abnormal phototransduction and behavior (Littleton et al., 1998 PMID: 9728921). Therefore, we cannot conclude that the ERG responses and phototaxis behaviors in NSF1 mutant flies resulted from disruption of Hdc axonal transport or the defective synaptic vesicle release. We therefore rather do not show these results in our manuscript.

Additional concerns:

Why does hHdc restore axonal histamine but not the ERG? Seems like partial rescue since phototaxis was partially restored.

Thanks for your concern. We confirm the phototaxis data and find that hHdc cannot restore the axonal histamine level or the ERG response. The hHdc was expressed in both the cell body and the synaptic terminal. The total level of histamine was not altered (Figure EV1C), thus the histamine in the synaptic terminal was reduced and not enough to maintain normal phototransduction.

The panels to the right in Fig. 2A seem to be reversed-the strep-tagged syt1 seems to be pulled down by HA whereas the HA-tagged is not.

We apologize for the mistake. The mislabeled Figure 2A has been replaced with correct one.

There are also a number of misstatements in the introduction and elsewhere:

Dense core vesicles are actually most abundant in axons, although not always presynaptic.

All of Tourettes is not caused by a mutation in Hdc.

DOPA does not accelerate PD-it is just associated with side effects of treatment.

We thank the reviewer for pointing out these mistakes. We apologize for the misstatements in the article. The correct ones have been highlighted in the article.

Referee #3:

Summary:

In this paper, Peng and Wang investigate the mechanisms of histamine synthesis and packaging into synaptic vesicles in *Drosophila* photoreceptors and the restriction of this process to axon terminals.

The authors showed the importance of local synthesis of histamine and its role as an important contributor to the synthesis of Hdc in the axon terminals for a normal synaptic transmission, hence the processing of visual input. They then explored their hypotheses on the localization of Hdc in synaptic terminals and discovered that the mature proteins were transported to the axon terminals via synaptic vesicles by proving their physical interaction. Furthermore, they identified NSF1 as the protein to which Hdc binds for its transport via synaptic vesicles. Additionally, they explored the quality control mechanisms for correct Hdc localization. After confirming that mislocalized Hdc is degraded through the ubiquitin-proteasome system, they performed a gene screening for E3 ubiquitin ligases, one of the key factors in the ubiquitin-proteasome system, and found that Purity of essence is required for Hdc degradation. Finally, they showed that knocking down these quality control mechanisms leads to the mislocalization of Hdc and hence histamine, which can cause retinal degradation, together illustrating the significance of the process of transporting Hdc to axon terminals.

This study, through a well-structured, wide variety of experiments, advances our understanding of histamine synthesis and localization mechanisms within *Drosophila* photoreceptors, shedding light on fundamental processes underlying synaptic transmission and emphasizing the crucial role of local histamine synthesis in visual processing.

Major Comments:

1) The experiments with Hdc mutant fly lines lack explicit reference to the mutation throughout the main text, and the implications of this mutation on protein function remain unexplained. While images indicate alterations in Hdc levels and histamine synthesis, further clarification on the effects of the mutation and a reference is needed.

We thank the reviewer for this suggestion. We Hdc mutant used throughout of the manuscript is the *Hdc*^{P217} mutant, generated by W. L. Pak's lab (Burg et al, 1993; PMID: 8096176). The *Hdc*^{P217} mutant has been showing a significantly reduced Hdc protein levels and activity in multiple studies including our previous study (Han et al. 2022 PMID: 35229720). We now add both references on page 5 high-lighted. We also defined the Hdc mutant as "Hdc mutant flies (*Hdc*^{P217})" in the legend of Figure 1.

2) The NSF1 and Hdc coimmunoprecipitation experiments were done in cell culture and are well explained. Moreover, they confirm the NSF knock-down experiments. However, although the fly lines are available and used in the experiments, an explanation is lacking why it would not be feasible to do the experiments in flies.

As suggested, we performed the co-IP experiments using fly tissues expressing both NSF1-GFP and Hdc-mCherry (*trp-Hdc-mCherry/culd-NSF1-GFP* flies). As a result, the

Hdc-mCherry efficiently interacted with NSF1-GFP but not with GFP, confirming the interaction of NSF1 and Hdc *in vivo* (reviewer figure below). We generated a new Figure EV3F, and added this result to the results section (Page 7 high-lighted).

Minor Comments:

- 1) Results, second paragraph: 'By contrast, GFP expressed under via the trp promoter localized to the retina, lamina, and medulla'. 'via' must go. There are a couple of other typos in the ms as well.
 - 2) A minor issue pertains to the naming of Hdc mutants in figures. Utilizing mutant names, such as HdcP217, rather than solely 'Hdc,' could enhance clarity.
 - 3) In Figure S3, D and E can be rearranged for a cleaner separation.
- We apologize for these mistakes. "via" is removed and the new sentence is highlighted. We agree that *Hdc*^{P217} is better than solely 'Hdc'. And 'Hdc' in the article is replaced with *Hdc*^{P217}.

Non-essential Suggestion:

- 1) The conclusory statements in several paragraphs, including the abstract and discussion appear somewhat tentative and not strong enough despite the robust evidence presented.
- Thank you so much for your suggestions, and we do appreciate your support. We changed several places in the discussion with vigorous words

Conclusion:

In conclusion, Peng and Wang's study advances our understanding of histamine synthesis and localization mechanisms in *Drosophila* photoreceptors, providing valuable insights into their importance in synaptic transmission, thus, visual processing. While the concerns raised are relatively minor, addressing them would further enhance the clarity and impact of their findings, ensuring the article reaches its full potential.

Dear Prof. Wang,

Congratulations on a great revision! Overall, the referees have been positive. However all three have a few more suggestions that we ask you to (non-experimentally if you choose) address in a new revision, including to please have your manuscript thoroughly edited. When you submit your revised version, please also take care of the following editorial items, and add this also to your point response:

1. Please remove the figures from the main manuscript file.
2. Please reduce the number of keywords to 5.
3. Beijing Municipal Commission of Science and Technology Commission, and Chinese Ministry of Science and Technology are missing in eJP. Are grant numbers available? If so please add these to your online account for this manuscript.
4. Please remove the author contribution section from the main manuscript.
5. Please review our new policy on conflict of interests on the EMBO author guide website and update the title of this section to: Disclosure and competing interests statement.
6. We include a synopsis of the paper (see <http://emboj.embopress.org/>). Please provide me with a general summary statement and 3-5 bullet points that capture the key findings of the paper.
7. We also need a summary figure for the synopsis. The size should be 550 wide by 200-440 high (pixels). You can also use something from the figures if that is easier.
8. Please clarify whether the callouts for Tables S1 and S2 should actually be for Tables and 2.
9. Please note that the figures EV 4-5 are mislabeled as figures EV 5-6 in the manuscript. This needs to be rectified.
10. Please define the annotated p values ****/**/* as well as provide the exact p-values for the same in the legend of figure 1c, e-f; 2c; 3a, d; 4a-f; 5b, d; EV 1b-c; EV 2d; EV 3d, h; EV 4c; EV 5b-c; as appropriate.
11. Please indicate the statistical test used for data analysis in the legends of figures 1c, e-f; 2c; 3a, d; 4a-f; 5b, d; EV 1b-c; EV 2a, d; EV 3c-d, g-h; EV 4a, c; EV 5a-c.
12. Please note that information related to n is missing in the legends of figures 4a, c-e.
13. Please note that the error bars are not defined in the legends of figures 1c, e-f; 2c; 3a, d; 4a-f; 5b, d; EV 1b-c; EV 2a, d; EV 3c-d, g-h; EV 4a, c; EV 5a-c.

Thank you for the opportunity to consider your work for publication, I look forward to your revision!

Warm regards,
Kelly

Yours sincerely,

Kelly M Anderson, PhD
Editor, The EMBO Journal
k.anderson@embojournal.org

Further information is available in our Guide For Authors: <https://www.embopress.org/page/journal/14602075/>

authorguide

Referee #1:

In the revised manuscript the authors addressed all issues previously raised by the reviewers.

My only remaining minor concern relates to some typos in the manuscript. Examples are listed below. Hopefully the publication fees will provide for some editorial assistance in that regard. Other than that, this interesting manuscript is ready for publication.

p. 4: Mutations in the Hdc gene results in symptoms of Tourette syndrome in both ...

should be: Mutations ... result...

p.4: In contrast, GFP expressed under the trp promoter localized to the retina, lamina, and medulla (Fig. EV1B).

Perhaps better to say:

In contrast, GFP expressed under control of the trp promoter localized to the retina, and photoreceptor axons in lamina and medulla (Fig. EV1B).

p. 7: mCherry-tagged Hdc along with MYC-tagged NSF1G274E, a mutation that disrupt ATPase activity of NSF1, in S2 cells.

should be: a mutation that disrupts...

p.7: Importantly, we found that the NSF1 G274E mutation has no effects on the binding ability of NSF1 and Hdc, did not affect the binding ability of NSF1 and Hdc, suggesting that NSF1's role in mediating axonal trafficking of Hdc is independent....

looks like part of the sentence is duplicated!

Referee #2:

The authors have responded to all of the concerns with explanation and where warranted, additional data. However, a few minor issues remain. In the Introduction, LDCVs are not located in SVs?! VMAT2 localizes to both LDCVs and SVs. And VMAT2 was cloned due to its ability to protect against MPP+ (Liu et al. Cell 1992), with the additional citations confirming and extending this original observation.

They did not understand the suggestion for the IP (Fig. 3b)-would be better to omit the Hdc-mCh (used for the IP) and make sure NSF does not come down.

The effect of NSF RNAi on histamine levels is a great addition, and it is fine to omit the effects on ERG and phototaxis given the role of NSF in transmitter release.

The only major concern that remains involves the discrepancy between rescue of histamine and the loss of ERG/major impairment in phototaxis with human Hdc. The authors make a reasonable point, that the total histamine does not change, and with more in the cell body, there should be less in the terminals. But the staining for histamine in the terminals appears very strong. Consistent with their hypothesis, are these cells just sick? They show evidence of retinal degeneration in Figure 5, but presumably the terminals degenerate as well? They ought to consider this possibility.

Referee #3:

The authors have adequately responded to all reviewer comments and clarified the ms accordingly. Just one newly added sentence confuses me a bit: (Line 44) 'Subsequently, they are transported and stored in large dense core vesicles (LDCVs) located within SVs at axon terminals.' This sounds as if LDCVs are located within SVs. Apart from this, the ms can now go to press as is.

Referee #1:

In the revised manuscript the authors addressed all issues previously raised by the reviewers.

My only remaining minor concern relates to some typos in the manuscript. Examples are listed below. Hopefully the publication fees will provide for some editorial assistance in that regard. Other than that, this interesting manuscript is ready for publication.

p. 4: Mutations in the Hdc gene results in symptoms of Tourette syndrome in both ...

should be: Mutations ... result..

The "results" is corrected to "result"

p.4: In contrast, GFP expressed under the trp promoter localized to the retina, lamina, and medulla (Fig. EV1B).

Perhaps better to say:

In contrast, GFP expressed under control of the trp promoter localized to the retina, and photoreceptor axons in lamina and medulla (Fig. EV1B).

We changed the sentence to: "In contrast, GFP expressed under control of the trp promoter localized to the retina, and photoreceptor axons in lamina and medulla (Fig. EV1B)." as suggested on P4.

p. 7: mCherry-tagged Hdc along with MYC-tagged NSF1G274E, a mutation that disrupt ATPase activity of NSF1, in S2 cells.

should be: a mutation that disrupts...

The "disrupt" is corrected to "disrupts"

p.7: Importantly, we found that the NSF1 G274E mutation has no effects on the binding ability of NSF1 and Hdc, did not affect the binding ability of NSF1 and Hdc, suggesting that NSF1's role in mediating axonal trafficking of Hdc is independent....

looks like part of the sentence is duplicated!

We deleted the duplicated sentence "did not affect the binding ability of NSF1 and Hdc" on P7.

Thanks for the reviewer for finding these typos. All mistakes listed above have been corrected. We also hope that publication fees will provide for reviewers and editors for assistance.

Referee #2:

The authors have responded to all of the concerns with explanation and where warranted, additional data. However, a few minor issues remain. In the Introduction, LDCVs are not located in SVs?! VMAT2 localizes to both LDCVs and SVs. And VMAT2 was cloned due

to its ability to protect against MPP+ (Liu et al. Cell 1992), with the additional citations confirming and extending this original observation.

We thank the reviewer for pointing out this misleading sentence. We changed the sentence to “Subsequently, they are transported and stored in large dense core vesicles (LDCVs) and SVs by vesicular monoamine transporter 2 at axon terminals”. We also added the references (Liu et al. Cell 1992, PMID: 1505023) on P3.

They did not understand the suggestion for the IP (Fig. 3b)-would be better to omit the Hdc-mCh (used for the IP) and make sure NSF does not come down.

We agree that an IP with NSF1 only is proper control to exclude the possibility of NSF1-Myc binding with RFP beads. Although we did not include this control in the 3b, in Fig. 3c, the NSF1-Myc is co-IPed with Hdc-mCherry and N-Hdc-mCherry but not with Hdc-DN-mCherry confirmed the physical interaction between NSF1 and mCherry, and excluded the possibility the NSF1 directly binding with RFP beads.

The effect of NSF RNAi on histamine levels is a great addition, and it is fine to omit the effects on ERG and phototaxis given the role of NSF in transmitter release. The only major concern that remains involves the discrepancy between rescue of histamine and the loss of ERG/major impairment in phototaxis with human Hdc. The authors make a reasonable point, that the total histamine does not change, and with more in the cell body, there should be less in the terminals. But the staining for histamine in the terminals appears very strong. Consistent with their hypothesis, are these cells just sick? They show evidence of retinal degeneration in Figure 5, but presumably the terminals degenerate as well? They ought to consider this possibility.

Thanks for your suggestions. We use histamine immunostaining to quantify the fluorescence ratio between the retina and the lamina within a picture. So, the fluorescence intensity between the pictures is not comparable due to different bright adjustments. It is precise to quantify histamine amount using LC-MS as shown. Therefore, we proposed that the relatively low content in the lamina of hHdc expressing flies.

We demonstrated that histamine accumulation in the retina caused retinal degeneration. As reviewers pointed out it is interesting to see if this histamine accumulation also leads to axon degeneration. However, in this manuscript, we did not follow the direction of axon damage. It might be a good direction for the future study. Thanks again for the advice on the manuscript.

Referee #3:

The authors have adequately responded to all reviewer comments and clarified the ms accordingly. Just one newly added sentence confuses me a bit: (Line 44) 'Subsequently, they are transported and stored in large dense core vesicles (LDCVs) located within SVs at axon terminals.' This sounds as if LDCVs are located within SVs.

Apart from this, the ms can now go to press as is.

We thank the reviewer for pointing out this misleading sentence. We changed the sentence to "Subsequently, they are transported and stored in large dense core vesicles (LDCVs) and SVs by vesicular monoamine transporter 2 at axon terminals".

Dear Prof. Wang,

Congratulations on an excellent manuscript, I am pleased to inform you that your manuscript has been accepted for publication in The EMBO Journal. Thank you for your comprehensive response to the referee concerns and for providing detailed source data. It has been a pleasure to work with you to get this to the acceptance stage.

I will begin the final checks on your manuscript before submitting to the publisher next week. Once at the publisher, it will take about 3 weeks for your manuscript to be published online. As a reminder, the entire review process, including referee concerns and your point-by-point response, will be available to readers.

I will be in touch throughout the final editorial process until publication. In the meantime, I hope you find time to celebrate!

Warm wishes,
Kelly

Kelly M Anderson, PhD
Editor, The EMBO Journal
k.anderson@embojournal.org
